# Cost-effective micro-CT system for non-destructive testing of titanium 3D printed medical components

**Santiago Fabian Cobos**[1], **Christopher James Norley**[2], **Steven Ingo Pollmann**[2], **David Wayne Holdsworth**[1,2]*

**1** Department of Medical Biophysics, Schulich School of Medicine & Dentistry, The University of Western Ontario, London, Ontario, Canada, **2** Imaging Research Laboratories, Robarts Research Institute, The University of Western Ontario, London, Ontario, Canada

* david.holdsworth@robarts.ca

## Abstract

Micro-CT imaging can be used as an effective method for non-destructive testing (NDT) of metal 3D printed parts–including titanium biomedical components fabricated using laser powder-bed-fusion (LPBF). Unfortunately, the cost of commercially available micro-CT scanners renders routine NDT for biomedical applications prohibitively expensive. This study describes the design, manufacturing, and implementation of a cost-effective scanner tailored for NDT of medium-size titanium 3D printed biomedical components. The main elements of the scanner; which include a low-energy (80 kVp) portable x-ray unit, and a low-cost lens-coupled detector; can be acquired with a budget less than $ 11000 USD. The low-cost detector system uses a rare-earth phosphor screen, lens-coupled to a dSLR camera (Nikon D800) in a front-lit tilted configuration. This strategy takes advantage of the improved light-sensitivity of modern full-frame CMOS camera sensors and minimizes source-to-detector distance to maximize x-ray flux. The imaging performance of the system is characterized using a comprehensive CT quality-assurance phantom, and two titanium 3D-printed test specimens. Results show that the cost-effective scanner can survey the porosity and cracks in titanium parts with thicknesses of up to 13 mm of solid metal. Quantitatively, the scanner produced geometrically stable reconstructions, with a voxel size of 118 μm, and noise levels under 55 HU. The cost-effective scanner was able to estimate the porosity of a 17 mm diameter titanium 3D-printed cylindrical lattice structure, with a 0.3% relative error. The proposed scanner will facilitate the implementation of titanium LPBF-printed components for biomedical applications by incorporating routine cost-effective NDT as part of the process control and validation steps of medical-device quality-management systems. By reducing the cost of the x-ray detector and shielding, the scan cost will be commensurate with the overall cost of the validated component.

## I. Introduction

Industrial computed tomography, for the purpose of non-destructive testing (NDT) of metal components, has developed to a mature technology of "turn-key" systems; this evolution has

**Data Availability Statement:** All raw projection images and reconstructed volumes are available from the Federated Research Data Repository

(FRDR) database (accession number(s) https://doi.org/10.20383/103.0603).

**Funding:** D.H. FDN 148474 (Canadian Institutes of Health Research) https://cihr-irsc.gc.ca/ NO D.H. RE-077-66 (Ontario Research Fund Research Excellence) https://www.ontario.ca/page/ontario-research-fund-research-excellence NO.

**Competing interests:** The authors have declared that no competing interests exist.

been thoroughly described in the literature [1, 2]. CT systems have naturally developed to fit applications that require resolution of features as small as 10 μm [1]. These micro resolutions are typical of industrial computed tomography scanners, which have strived to achieve high-resolution instrumentation through dense metal structures. Bench-top micro-CT scanners are widely used for (NDT) of parts in the automotive, aerospace, pharmaceutical, and medical industries [3–5]. In particular, micro-CT based NDT has been recently introduced as a volumetric measurement tool for additive manufacturing (AM) [6].

The main components of these scanners are the x-ray source, the x-ray detector, and a rotary stage that are all enclosed in an x-ray shielded cabinet [2, 7]. With prices ranging from 100,000 to over 1 million USD dollars, industrial micro-CT scanners for NDT are typically equipped with high-energy (160–450 kVp) x-ray tubes that require heavy x-ray shielding, additional hardware (e.g., power generators or cooling systems), and robust infrastructure to operate safely. For these high-energy systems, x-ray filtration is employed to modify the x-ray spectra, in order to scan samples made of various materials with a wide range of radiopacity (i.e., plastics, ceramics, and metals). Additionally, these scanners are equipped with large-area digital flat-panel x-ray detectors that allow for scanning of medium-to-large sized objects, which are characteristic of the automotive and aerospace industry [2–4, 8]. For applications where the cost of analysis must be commensurate with the cost of the part, these NDT tools become prohibitively expensive for routine use [8]. For example, in the medical industry, the cost of scanning a $300 part should not exceed more than approximately 10% of its cost, or $30.

Scanning costs can be reduced using micro-CT scanners designed with task-specific, x-ray tube and detector combinations [3]. For example, de Oliveira et al. [9] have described a low-cost (~ $35,000) scanner for small-sized specimens (< 100 mm) using a low-energy, x-ray tube in a fan-beam configuration, and an x-ray detector made in-house comprised of a series of commercial photodiode arrays. Their low-cost detector had only one row of 512 (0.3 x 0.6 mm) photodiodes, which constrained the reconstruction volume of the scanner to a single slice. To avoid this limitation, industrial and commercially available micro-CT scanners typically use a cone-beam geometry and two-dimensional flat-panel detectors which can be very expensive. Dramatically reducing the cost of a medium-sized high-definition detector is a very effective strategy for reducing the overall system cost and thus making routine NDT of medical components feasible.

Micro-CT is an excellent NDT tool to investigate the internal integrity (e.g., porosity and cracks) of a part, as well as its dimensional accuracy [10]. It should be emphasized that routine non-destructive testing of medical components should not have the objective of producing volumetric data for high-precision metrology of the parts, but rather used for an assessment of critical defects with the potential of affecting the mechanical properties of the component [11]. This assessment should include the size, location, and distribution of the defects [6, 12]. In the case of titanium medical components, the critical defect size might be 0.5 mm, requiring a spatial resolution for the CT volume of around 0.2 mm in order to visualize any mechanically critical defects [11, 12]. Additionally, the scan must be completed as fast as possible, but must produce sufficient image quality to identify and measure what is required [5]. This can be achieved through the use of flat-panel detectors with sufficiently-small pixel pitch, or the use of geometric magnification of the sample. Lens-coupled detector systems have been investigated in the past as an alternative, but have lacked the necessary light sensitivity or field-of-view, as they have been limited to small-scale imaging applications [13–16]. Recent advancements in full-frame CMOS imaging technology have shown promise in reaching the light sensitivity, field-of-view size, and spatial resolution required for computed tomography [17]. Panna et al. [18] have demonstrated the use of consumer-grade, digital cameras that are lens-

coupled to x-ray phosphor screens as a viable way of constructing high-definition medium-sized x-ray detectors. They also demonstrated that by placing the digital camera in front of the phosphor screen, in a front-lit tilted configuration, these lens-coupled detectors are less susceptible to diffusion and attenuation of light in the phosphor, and are able to perform as well as modern digital flat-panel detectors.

This study describes the first implementation of a low-energy (80 kVp) micro-CT system equipped with a medium-sized front-lit lens-coupled detector, as a cost-effective alternative for NDT of medium-sized medical components such as: cranial plates, interbody fusion devices, acetabular cups, maxillofacial implants, customized fixation plates, cervical cages, etc. The necessary hardware and software required to acquire high-quality micro-CT data is described, and the performance of the scanner is characterized using a comprehensive quality assurance phantom [19]. Finally, the use of this novel device to inspect two titanium 3D-printed test objects is described–one to evaluate the performance of the system while imaging highly-porous lattice structures, and the other to characterize x-ray penetration limits in this commonly-used medical-grade alloy.

## II. Methods

### A. System geometry description

The basic components of a micro-CT scanner are an x-ray tube, a rotational stage, and an x-ray detector. These components in the low-cost CT system were enclosed in a frame constructed using off-the-shelf 20 x 20 mm aluminum extrusion profiles and connectors. This 1040 x 500 x 540 mm enclosure included opaque non-reflective panels that create a light-tight environment inside the frame, preventing any external light from interfering with the lens-coupled detector system. All the hardware, including a portable x-ray source, could be packaged into a large suitcase with an estimated maximum weight of 100 kg. One side of the frame lacked a solid panel, but instead had an opaque, non-reflective fabric to facilitate access to the rotary stage for specimen loading. The enclosure and its internal components were placed inside a lead-shielded procedure room that prevented operator exposure to x-ray radiation.

Fig 1 describes the schematics of the frame and the internal components of the low-cost, micro-CT system. The configuration of the x-ray source, phosphor screen, and the camera-lens assembly is similar to the tilted-screen front-lit configuration described by Panna et al. [18] with the addition of a 2 mm thick lead plate placed in between the rotary stage and the camera to prevent direct interactions between the camera CMOS sensor and any primary or scattered x-rays. The camera-lens assembly was mounted to the frame with a sliding gantry plate that allowed camera-to-screen adjustments. The phosphor screen was secured to one end of the C-beam using an acrylic sheet. Underneath the acrylic sheet, a custom 3D-printed hinge joint allowed for the adjustment of the phosphor-screen tilt. All of the other joints and connectors allowed partial disassembly of the system, as well as adjustments to various geometric parameters (source-to-object distance (SOD), object-to-detector distance (ODD), detector-screen tilt angle, and camera-to-screen distance) making the design of the low-cost micro-CT system versatile, portable, flexible, and easily upgradable.

### B. Internal components and experimental setup

The schematic in Fig 1, shows the configuration of the proposed system. The system is designed to include a portable x-ray source with a 33 μm focal spot capable of running at a maximum tube potential of 80 kVp and a maximum current of 0.5 mA (Sourceblock SR-80-500). The source is placed in a close-assembly (i.e., source-to-detector distance (SDD) of approximately 300 mm) to maximize x-ray flux at the detector plane. Unfortunately, due to

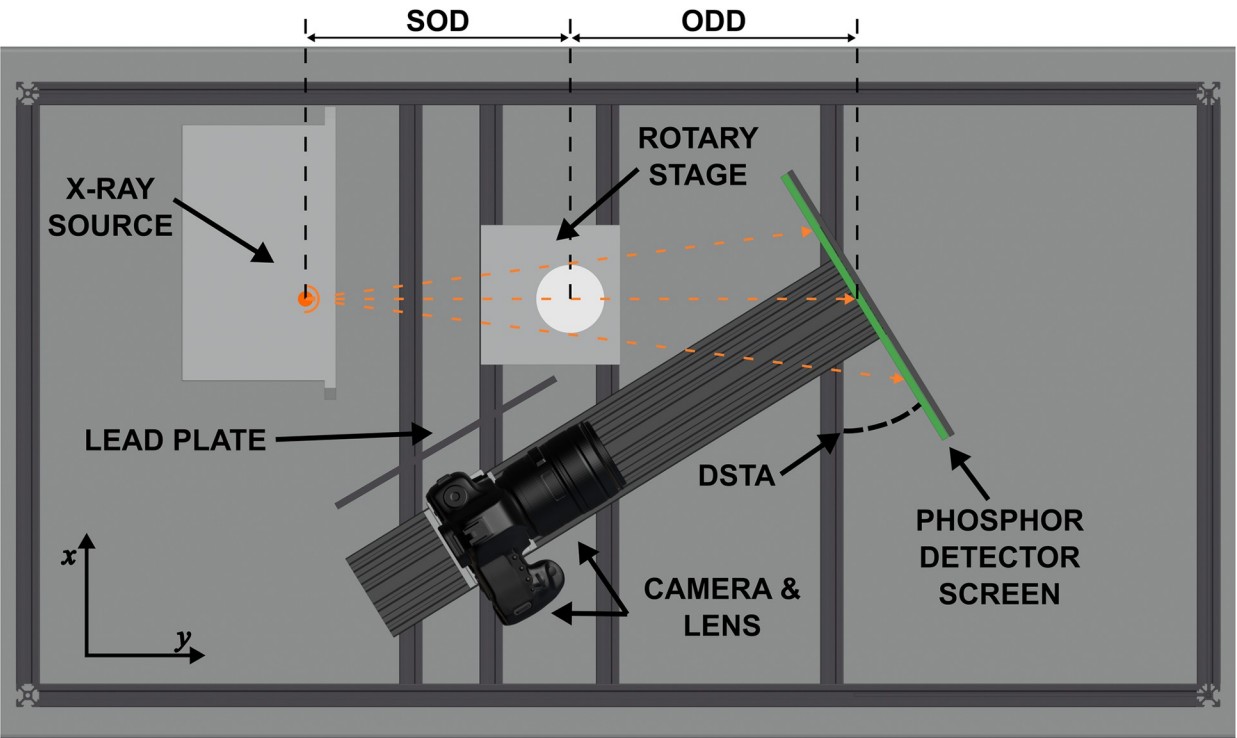

**(a)**

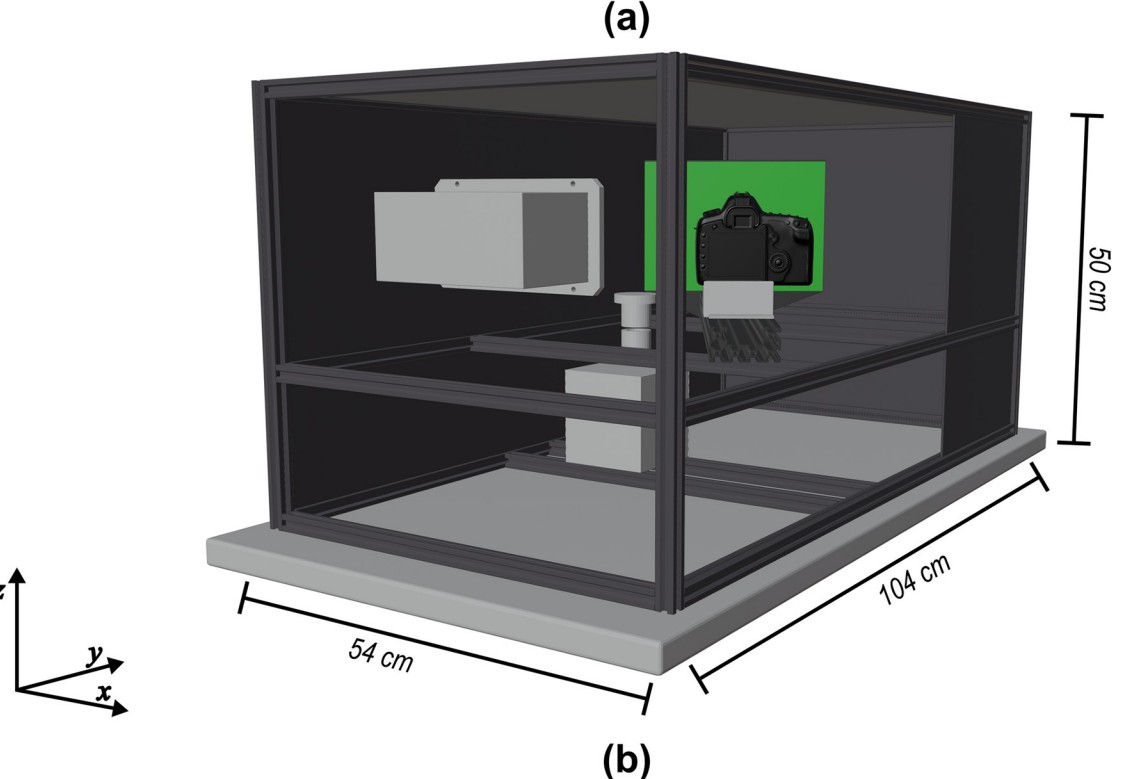

**(b)**

**Fig 1. Geometry of the micro-CT system.** (a) Top view depicting the arrangement of the internal components of the system: x-ray source, rotary stage, phosphor detector screen, detector-screen tilt angle (DSTA), lead plate, camera, lens, source-to-object distance (SOD), and object-to-detector distance (ODD). The orange dotted-lines represent the trajectory of the polychromatic, cone-shaped, x-ray beam. (b) Perspective view of the system showing the arrangement of the various 20 x 20 mm aluminum extrusion profiles. One of the opaque panels was removed from the 3D rendering to allow visualization of the internal components.

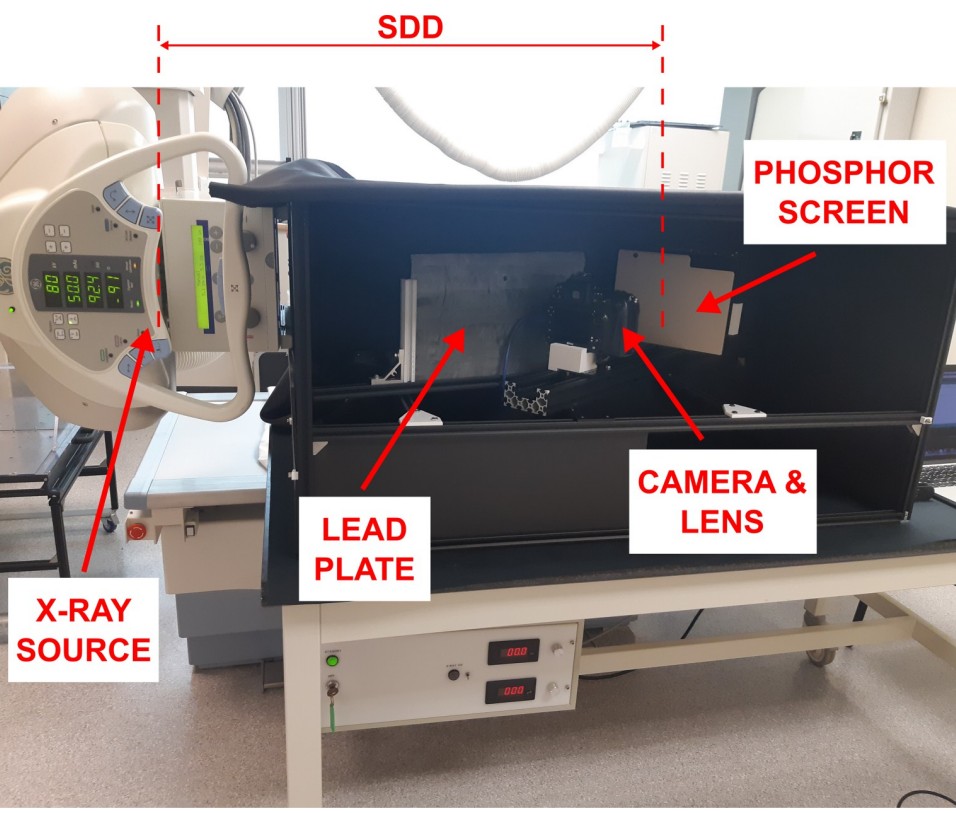

**Fig 2. Experimental setup using the general-purpose x-ray source.** Note that the rotary stage is not visible as it is hidden behind the lead plate. The source-to-detector distance (SDD) was 600 mm.

regulatory restrictions and reduced research capacity caused by the COVID-19 pandemic, the researchers were unable to operate that x-ray source and instead used a general-purpose ceiling-mounted x-ray source (Proteus XR/a, GE Medical Systems). The SDD for this configuration was 600 mm and images were captured at 80 kVp and 10 mA. Fig 2 shows the experimental setup using the general-purpose ceiling-mounted x-ray source. Note that the x-ray protocol used with the general-purpose x-ray source was adapted to match the exposure levels that could be achieved with the proposed portable x-ray source. Table 1 summarizes the differences between the two x-ray units and the exposure matching protocol settings.

**Table 1. X-ray source comparisons and exposure-level matching, x-ray protocols.**

|  | Portable x-ray Unit | | General Purpose x-ray Unit | |
| --- | --- | --- | --- | --- |
|  | Source-Ray/SR-80-500 | | GE Medical Systems/Proteus XR/a | |
|  | Unit | Protocol | Unit | Protocol |
| x-ray tube-potential (kVp) | 35–80 | 80 | 40–150 | 80 |
| x-ray tube-current (mA) | 0.5 | 0.5 | 10–400 | 10 |
| Focal spot size (mm) | 0.033 | 0.033 | 0.6 | 0.6 |
| Source-to-detector distance (mm) | - | 300 | - | 600 |
| Dose-rate* (R/h) | - | 344 | - | 1718 |
| Exposure time (s) | - | 4 | - | 0.8 |
| Tube-current exposure-time product (mAs) | - | 2 | - | 8 |

*Dose-rates were calculated using Rad Pro Calculator version 3.26

**Table 2. Summary of the estimated costs for the main components of the low-cost CT scanner–June 2021.**

| Component | Cost (USD) |
| --- | --- |
| Sourceblock SR-80-500 (portable x-ray unit) | $ 4495 |
| Nikon D800 dSLR camera | $ 3000 |
| Nikkor 35 mm f/1.4 lens | $ 1100 |
| Phosphor screen | $ 30 |
| Rotary stage and frame | $ 600 |
| X-ray shielding | $ 1500 |
| Total costs: | $ 10725 |

The rotational stage for the system was fabricated in-house and operated under a closed-loop stepper-motor control with < 5 μm precision. For each scan, 360 images were acquired at 1° angular increments. The stage was placed 300 mm from the x-ray source (source-to-object distance, SOD) and 300 mm from the phosphor screen central axis (object-to-detector distance, ODD). The magnification factor (M) of the system was calculated to be:

$$M = \frac{SDD}{SOD} = \frac{600 \ mm}{300 \ mm} = 0.5 \qquad (1)$$

The x-ray detector was comprised of a 180 x 240 mm $Gd_2O_2S$: Tb phosphor screen (Lanex Min-R 2000, Eastman Kodak Company), a fast high-quality lens (35 mm, f1.4 prime Nikkor), and a full-frame dSLR camera (Nikon D800). The camera-to-screen distance was set to 230 mm and the CMOS sensor focus was visually optimized prior data acquisition. Images were captured with the widest aperture setting (focal ratio of f/1.4) in order to maximize light collection. The camera exposure time was set to 4 s to match the exposure time required when using the proposed SR-800-500 x-ray source. This ensured that the appropriate amount of electronic noise (i.e., dark-field) was included during data acquisition and subsequent CT reconstruction. Finally, ISO sensitivity was set to 1600 ISO to cover the full dynamic range (14 bits) of the Nikon D800 camera.

Several private companies selling micro-CT, NDT devices were surveyed and reported scanner prices ranging from $ 260,000 for a small field-of-view scanner to over $ 750,000 for a large field-of-view scanner. Table 2 details the costs of all the off-the-shelf components of the proposed low-cost scanner. Due to the use of a relatively low-energy source (80 kVp), the calculated shielding requirements for routine use of the proposed system were significantly lower than other scanners operating at higher energies. For instance, reduction of the radiation measurements caused by primary x-rays behind the phosphor screen to the required 0.005 mSv/h can be achieved using an 8.4 mm lead plate. Additional shielding will be required to enclose the portable x-ray source, as the Sourceblock SR-80-500 unit is sold without x-ray protection. Fortunately, lead-based x-ray protection is not costly, unless its weight impacts other aspects of the design of the CT scanner.

## C. Geometric calibration and CT reconstruction

For these experiments, the Nikon D800 captured RAW 6208 x 4924 pixels 14-bit dynamic range images in 5:4 mode (30 by 24 mm sensor area). These images contained the pixel-by-pixel intensity values arranged in a 2 by 2 pixel Bayer mask (one red, two green, and one blue). The intensity values of each group of four pixels were averaged to generate a 3104 x 2462 image. Fig 3 shows an overview of the image acquisition and post-processing steps required prior to CT reconstruction.

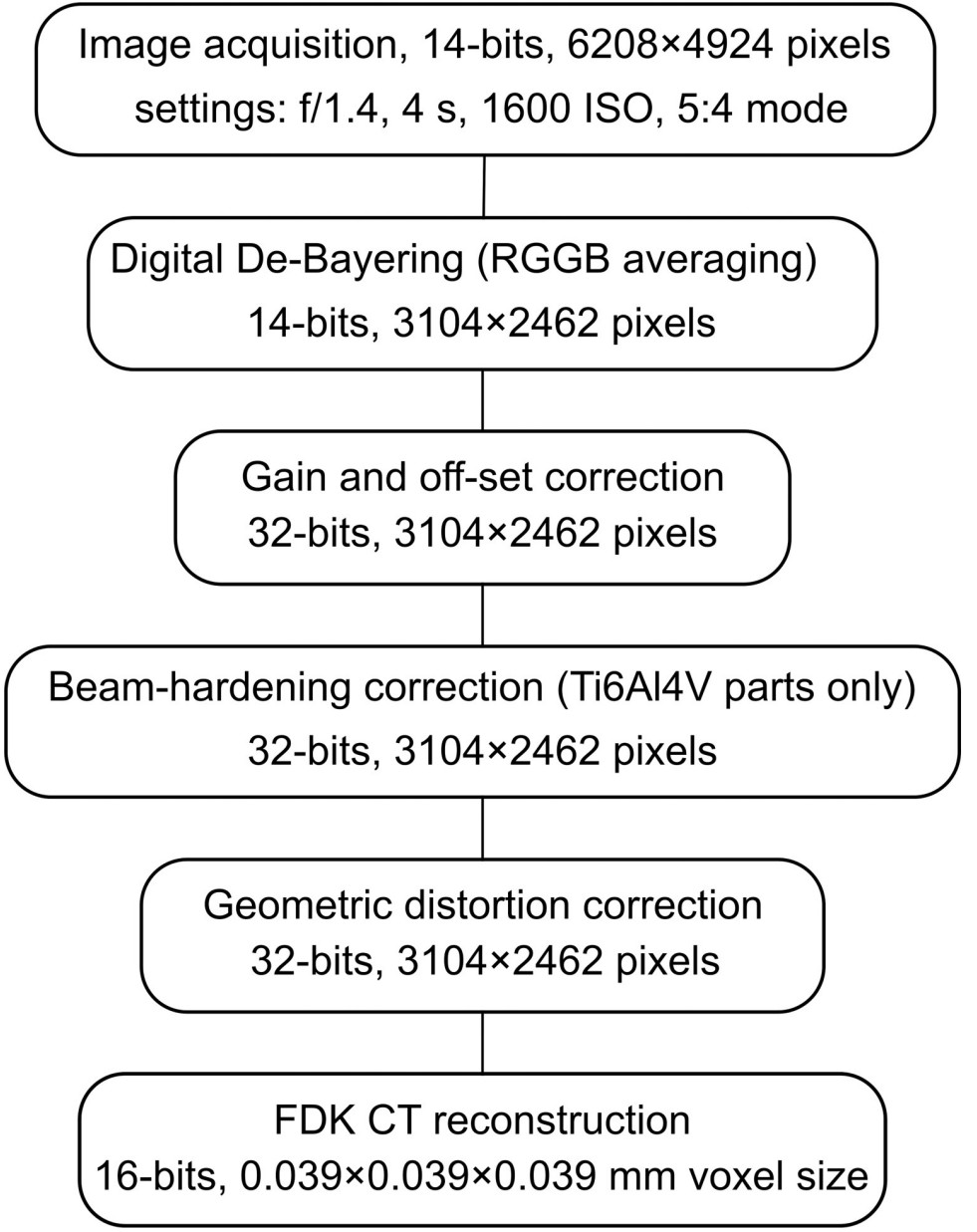

**Fig 3. Summary of image processing steps required between projection data acquisition and CT reconstruction.**

The image quality of any CT reconstruction depends on precise calibration of the imaging system. If the object space is not mapped correctly with the projection data, the reconstructions will have reduced spatial resolution, as well as image artefacts [20]. In the proposed system, this could be caused mainly by the geometric distortion of the object due to the tilted configuration of the phosphor, and by source-detector alignment errors. Fortunately, both geometric distortion and source-detector alignment are tractable problems.

Lens and perspective geometrical-distortion corrections are routinely implemented in photogrammetry and industrial metrology [21]. Additionally, it has been demonstrated that similar corrections can be applied in radiography to increase spatial resolution using two orthogonal views of a tilted x-ray detector [22]. For this study, geometric correction involved

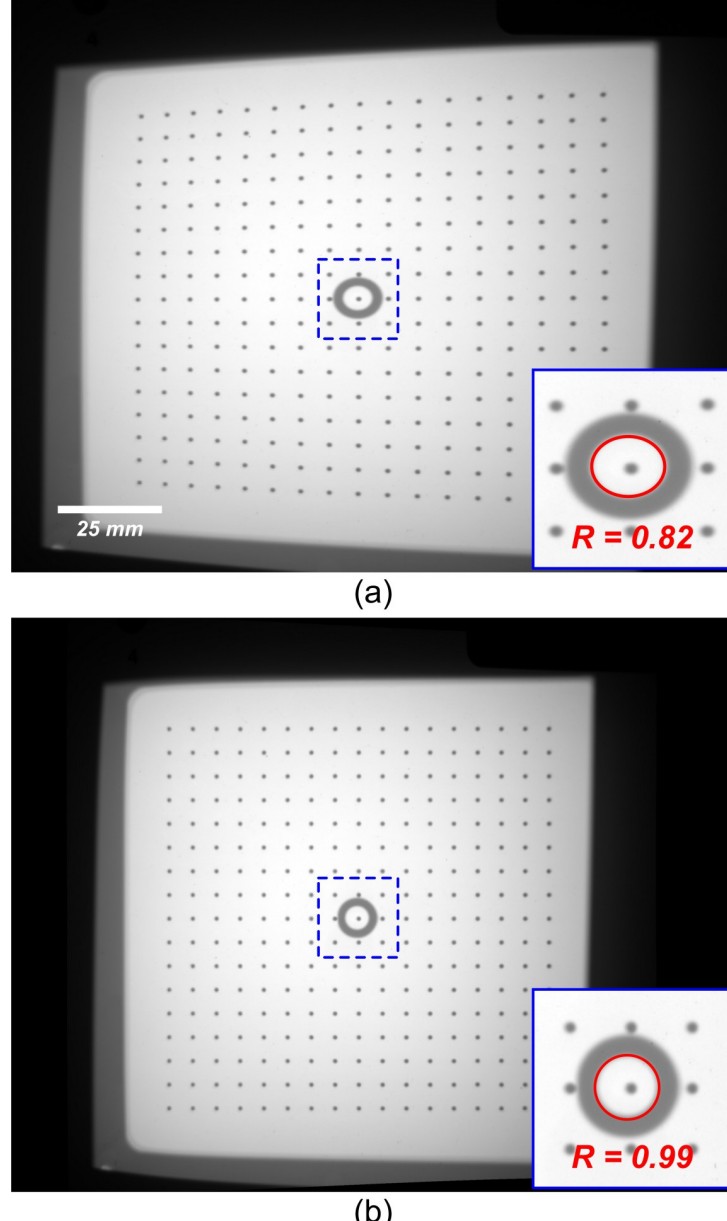

**Fig 4. Assessment of geometric distortion correction resulting from the tilted-detector configuration.** (a) shows the un-corrected raw data of the Cartesian calibration grid. (b) corrected image of the Cartesian calibration grid.

capturing a projection image of a precisely fabricated 80 x 80 mm grid containing 17 rows by 17 columns of 0.5 mm diameter high-contrast metal marker beads. The centroid of each marker bead was determined to subpixel accuracy and used to correct the image based on the a priori knowledge about the Cartesian grid [23]. Source and detector alignment was estimated using a ring-shaped high-contrast object that was placed 80 mm from the plane of the Cartesian grid toward the x-ray source [24]. Proper alignment of this ring and the central marker of the grid was visually confirmed before data acquisition (Fig 4). The accuracy of the correction was evaluated, primarily, by imaging a cylindrical phantom with a series of high-contrast metal beads placed on the outer surface of the cylinder parallel to its longitudinal axis. The

rotational path of the centroid of each marker was compared to their expected elliptical trajectory in one full rotation of the rotational stage. Additionally, the roundness scores of the ring-shaped marker were calculated according to Eq (2) and provided an additional two-dimensional evaluation of the geometric correction.

$$roundness\ (R) = \frac{4 \times area}{\pi \times major\ axis^2} \qquad (2)$$

For each data set, 360 projection images were acquired to perform a micro-CT reconstruction with a widely-employed Feldkamp, Davis and Kress (FDK) filtered-backprojection algorithm. Beam-hardening correction, following the protocol described by Edey et al., was applied for the titanium specimens only. For all reconstructions, CT numbers were linearly rescaled into Hounsfield (HU) units. The voxel spacing of the original reconstruction matrix was 39.3 x 39.3 x 39.3μm. Following a quantitative assessment of the spatial resolution of the system, described below, all reconstructions were spatially averaged to 118 x 118 x 118 μm to improve the signal-to-noise characteristics of the data.

## D. Phantoms

**a. Quality assurance phantom.** Fig 5 shows the three phantoms used in this study. The quantitative performance of the system was evaluated using a comprehensive micro-CT quality-assurance phantom (Fig 5A) designed and described by Du et al. [19]. The different sections of the QA phantom were used to assess spatial resolution, geometric accuracy, CT number accuracy, linearity, uniformity, and noise.

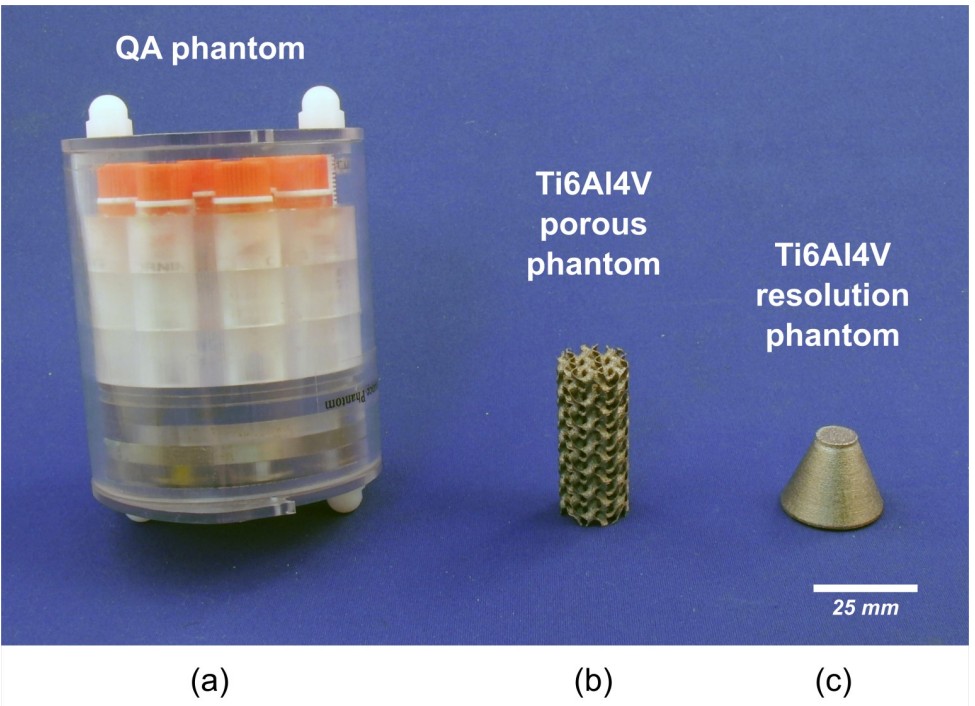

**Fig 5. Physical appearance of the imaged phantoms for this study.** (a) comprehensive quality assurance CT phantom, (b) porous, gyroid-based, cylindrical, titanium-alloy (Ti6Al4V) phantom, and (c) titanium-alloy (Ti6Al4V) resolution phantom.

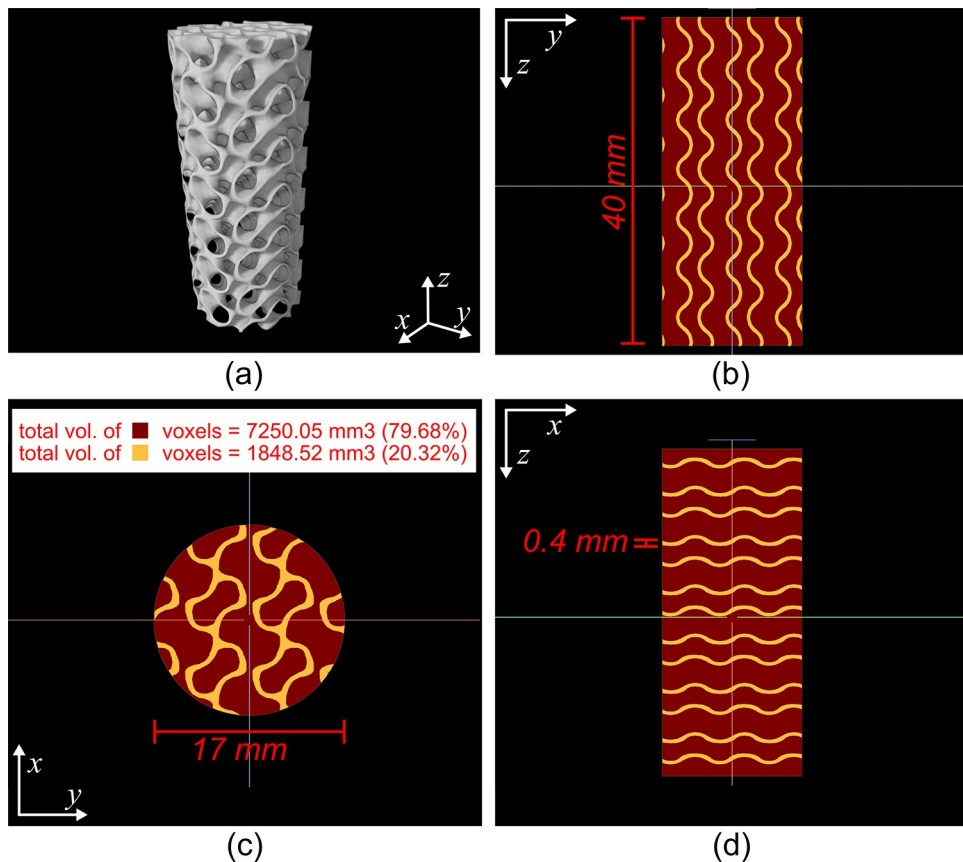

**Fig 6. Computer-aided design (CAD) of the porous gyroid-based cylindrical titanium-alloy (Ti6Al4V) scaffold.**
(a) perspective view of 3D rendering of the CAD. (b) Trans-coronal synthetic slice of the porous cylinder CAD. (c) Trans-sagittal synthetic slice of the porous cylinder CAD. (d) Transaxial synthetic slice of the porous cylinder CAD. (b),(c), and (d) are color coded to depict the titanium-alloy (yellow) and internal porosity (red) regions of the scaffold.

**b. Porous scaffold phantom.**   To assess the capability of the system to visualize internal voids in a commonly-used medical-grade alloy, two titanium (Ti6Al4V ELI-0406, Renishaw plc, United Kingdom, particle size 15–45 µm) test specimens were manufactured with a metal 3D printer (AM400, Renishaw plc, Wotton-under-Edge, United Kingdom) at the Additive Design in Surgical Solutions facility (ADEISS, London, Canada). The first titanium test specimen (Figs 5B and 6) was a porous cylindrical scaffold 17 mm in diameter and 40 mm in length. The internal porosity of the scaffold was designed in Blender (Version 2.79, blender.org, Amsterdam, Netherlands) using a 6 mm$^3$ sheet-based gyroid unit cell. The prescribed porosity of the porous scaffold was 79.68% and was achieved by modifying the wall thickness of the gyroid unit to 0.4 mm (Fig 6C and 6D).

**c. Titanium resolution phantom.**   The second titanium test specimen (Figs 5C and 7) was a truncated cone 17 mm high with a 25 mm diameter base, and a 10 mm diameter top. Incorporated into the design were a series of deliberate internal cavities or voids which, in cross-sectional view, were seen as: a set of five bar patterns (formed from alternating regions of titanium and air with nominal spacing varying from 1.66 to 10 lp/mm); a set of five circular air cavities with nominal diameters varying from 0.025 to 0.3 mm; and a set of five rectangular air cavities 3 mm wide with varying nominal thicknesses from 0.025 to 0.3 mm. These internal cavities extended from the top of the truncated cone to the base along the phantom's longitudinal axis. The top and the base of the truncated cone were two, 1 mm thick, disk-shaped

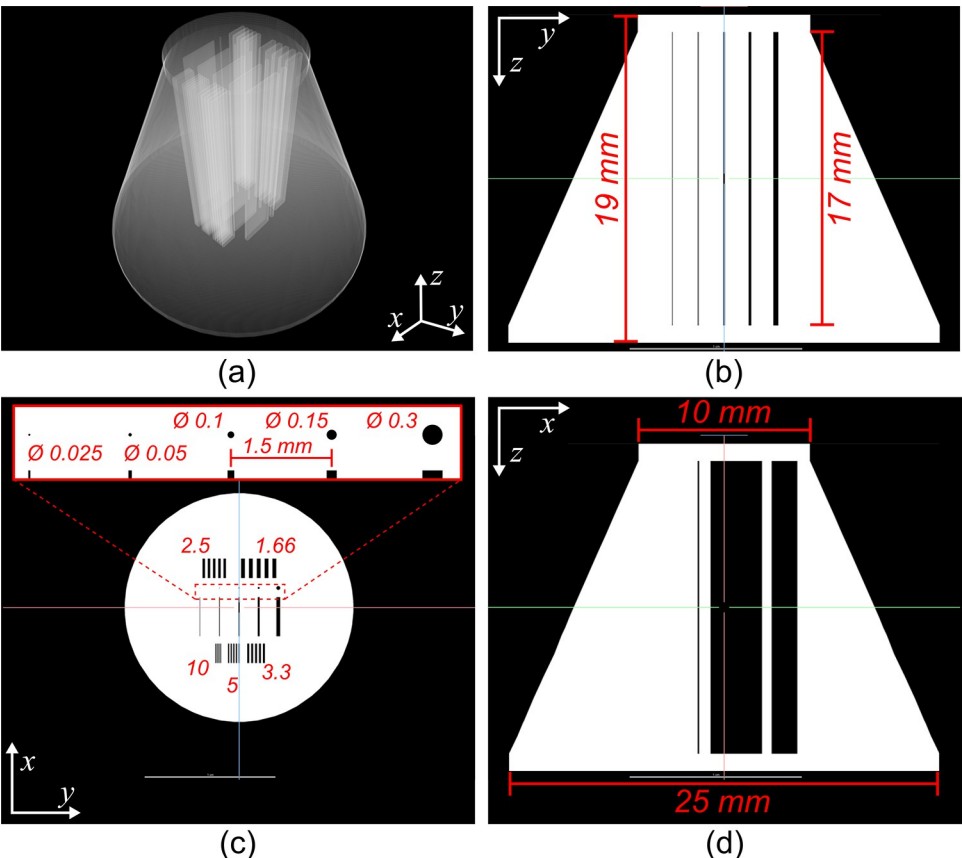

**Fig 7. Computer-aided design (CAD) of the resolution titanium-alloy (Ti6Al4V) phantom.** (a) translucent perspective view of 3D rendering of the phantom's CAD. (b) Trans-coronal synthetic slice of the phantom's CAD. (c) Trans-sagittal synthetic slice of the phantom's CAD depicting details of the internal voids. The bar patterns are described in lp mm$^{-1}$ all the other nominal measurements are shown in mm (d) Transaxial synthetic slice of the phantom's CAD. Note that the labelled measurements represent the prescribed nominal values; these values may not be achieved during the manufacturing of the part.

regions. Fig 7 shows a detailed schematic of this titanium resolution phantom that was designed to assess the ability of the low-cost micro-CT system to resolve features of various sizes inside a solid object as the cumulative x-ray path length across the material increases. The x-ray path length increases as the diameter of the cross-sectional area of the cone increases towards the base of the phantom.

This titanium resolution phantom was imaged using an advanced commercially available micro-CT scanner (Zeiss Xradia 410 Versa, Germany) at an ISO 17025:2017 certified facility (Surface Science Western, London, Canada). The scanning parameters were as follows: the phantom was rotated and scanned at 0.225˚ increments. After each rotational stage, the phantom was imaged twice. The X-ray unit operated at 150 kV and 10 W power with an exposure time of 12 s. The 1601 projection images were reconstructed with a voxel size of 12.68 µm. This Zeiss Xradia scan of the resolution phantom was performed to characterize the true size of the phantom's internal voids, post-manufacturing, and to evaluate their geometry from top to bottom.

**E. Data analysis.** The spatial resolution of the micro-CT scanner was quantitatively measured over a range of spatial frequencies, using two different methods to calculate the MTF of the system. The first method used the slanted-edge plate of the QA phantom and the algorithm

described by Judy [25]. The second method, described by Anam et al. [26], used a cross-sectional reconstruction of a 0.1 mm diameter copper wire that was wrapped around one of the plates of the QA phantom. The spatial resolution was qualitatively evaluated using a different plate of the QA phantom that included four spiral coils fabricated with alternating aluminum and Mylar sheets with various thicknesses of 150, 200, 300, and 500 μm–corresponding to 3.3, 2.5, 1.6, and 1 lp/mm, respectively. Using Droege's method [27], the modulation parameter was calculated using the standard deviation of pixel values in a region-of-interest (ROI) within the coils. A set of these modulation parameters were useful to verify the MTF calculation for the proposed system.

Geometric accuracy was determined using another plate within the QA phantom containing five beads placed at precisely known locations. The 3D sub-voxel weighted centroid of each bead was calculated using standard methods and the distances from each centroid to that of all the other neighboring beads were measured in voxels. The known distances between beads were used to calculate the true in-plane voxel size of the reconstructions.

The CT number evaluation and linearity of the system were measured using two other plates of the QA phantom designed for these purposes. The CT number of each of the eight materials was reported by measuring the mean grey-scale intensity in Hounsfield units (HU) within a 1.5 x 1.5 x 1.5 mm ROI placed in the center of each material sample. Linearity was determined by vials of iodinated contrast agent (Omnipaque 300, GE Healthcare, Oakville, ON), with concentrations varying from 0.9375 to 30 mg ml$^{-1}$. A linear-regression analysis was used to determine the relationship between the signal intensity and iodine concentration.

The uniformity of the system was qualitatively evaluated using the uniformity plate of the QA phantom and a radial signal profile taken through the center of a homogeneous region covering the central reconstructed slice. Ten slices were averaged to improve the noise characteristics of the data and allow proper visualization of any non-uniformities. This plate was also used to quantitatively report uniformity and noise using four ROIs located at the periphery of the plate as well as one ROI located in the center. The uniformity percent and the average of the measured standard deviations for each ROI is reported.

An ROI of the titanium porous scaffold was segmented from the beam-hardening-corrected reconstruction using Dragonfly software, version 2020.2 (Object Research Systems Inc, Montreal, Canada) and a standard full-width-at-half-maximum (FWHM) threshold in order to determine the volume of titanium excluding the surrounding air. The centroid of each slice, perpendicular to the longitudinal axis of the sample, was used to draw a circular mask fitting the bounding box of the titanium ROI. These circular masks were combined to generate a cylindrical ROI that represented 100% of the volume of the titanium scaffold with the surrounding air excluded. The percentage porosity of the reconstructed sample was then calculated using:

$$CT \ scaffold \ porosity = \left( 1 - \frac{titanium \ ROI \ volume}{cylindrical \ ROI \ volume} \right) \times 100 \qquad (3)$$

This CT-derived porosity fraction was compared with the gravimetrically-determined, porosity fraction of the 3D printed sample, which was determined by weighing the printed scaffold using a high-precision scale (BP3100P, Sartorius, Göttingen, Germany, 0.01 g accuracy) and comparing it with the mass of a solid titanium cylinder.

The x-ray path-length limit for titanium was measured using the titanium resolution phantom. This limit was determined by measuring the diameter of the phantom at the height of the cone where its internal features–the centrally located voids–were no longer visible in a lateral

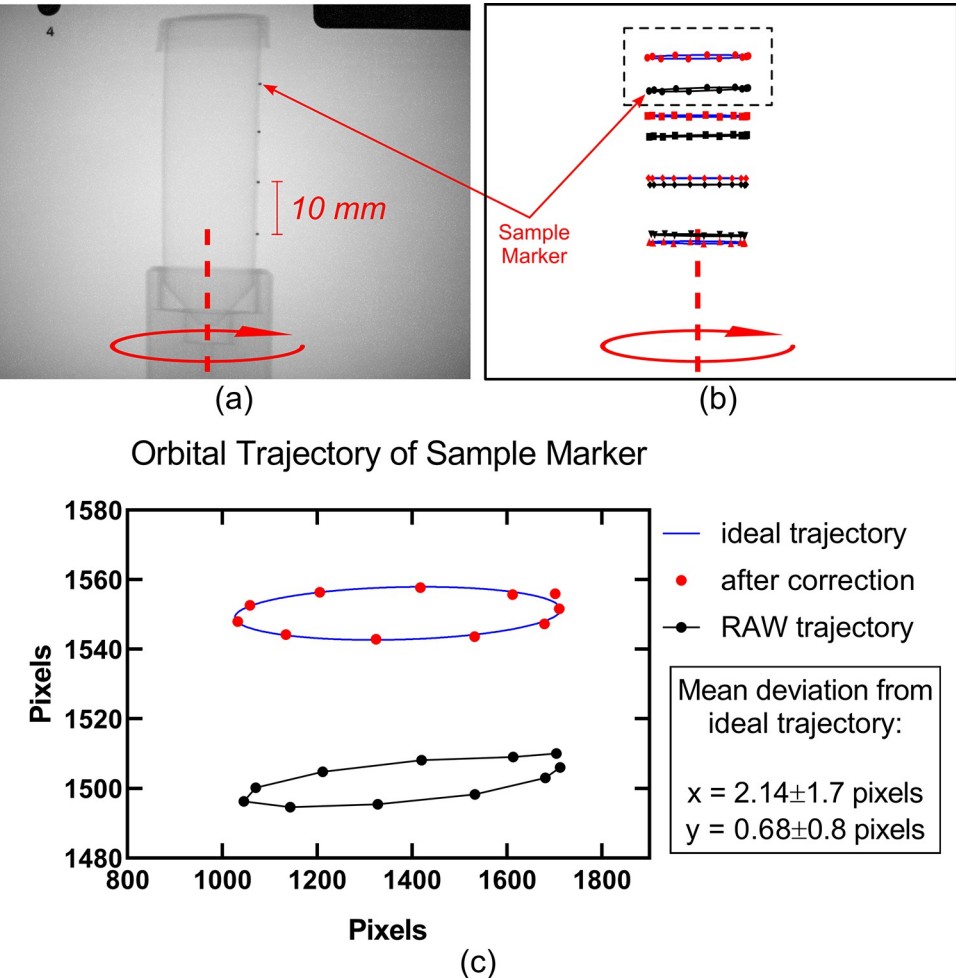

**Fig 8. Results of geometric calibration evaluation using a cylindrical phantom with four radiopaque marker beads.** (a) x-ray projection radiograph of the phantom; (b) uncorrected and corrected elliptical trajectories prescribed by the phantom's markers over the course of one full rotation of the rotary stage. (c) close-up of uncorrected and corrected elliptical trajectories of the top-most marker bead. The corrected position of the markers aligns with the ideal trajectory calculated using a sinusoidal best fit.

cross-sectional view. The phantom was also used to qualitatively evaluate the spatial resolution of the system when imaging highly radio-opaque materials.

## III. Results

### A. Geometric correction

The mean deviation from the ideal trajectory was measured (1.14 ± 1.34 pixels) showing that the correction rendered trajectories with deviations within the spatial resolution of the detector–which was mainly limited by penumbral blur of the focal-spot (~ 0.3 mm). Fig 8 shows a projection image of the phantom as well as the measured and expected elliptical trajectories of each marker.

### B. Imaging characteristics of the system

**a. Spatial resolution.** The spatial resolution of the system was quantitatively evaluated by calculating the pre-sampled MTF of the system using the reconstructed image of the slanted-edge plate of the QA phantom (Fig 9A). The resolution limit at the 10% level was reached at

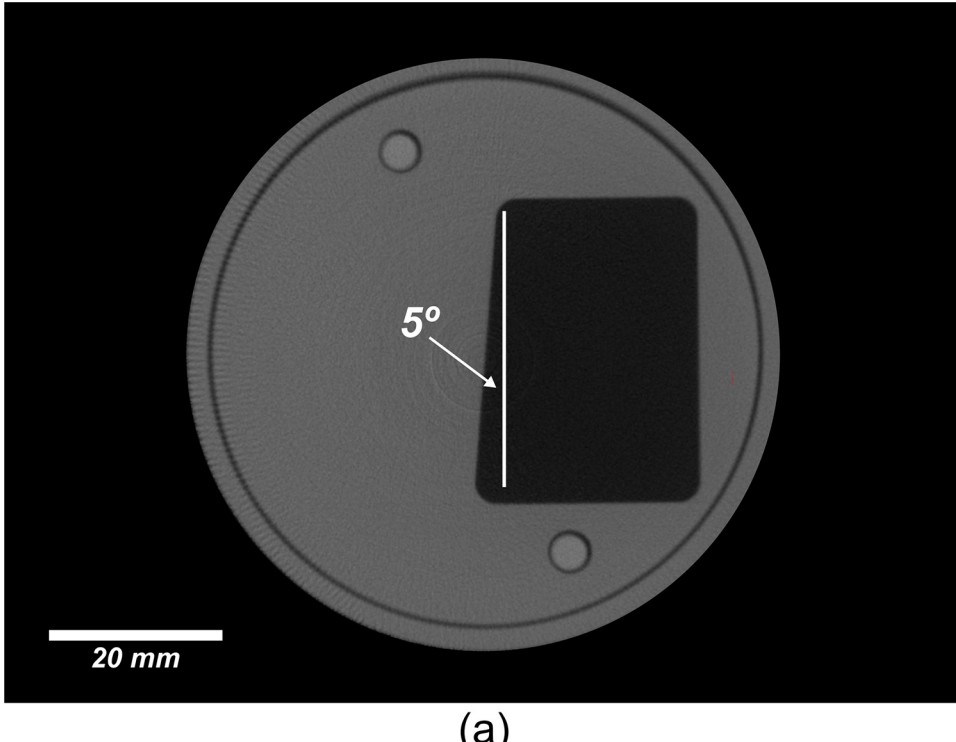

(a)

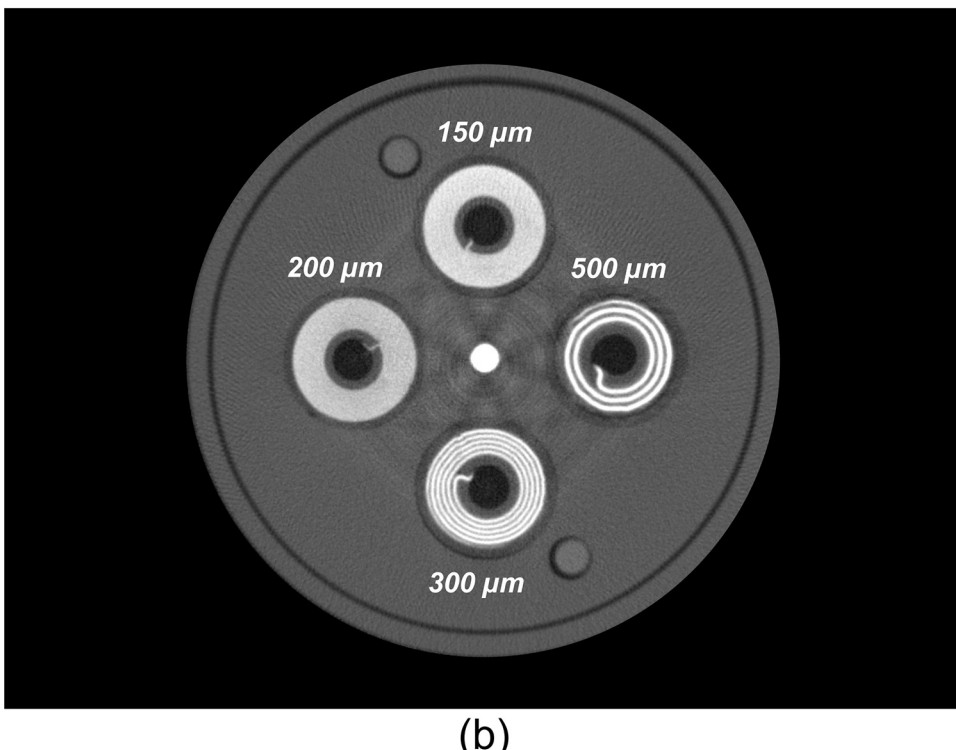

(b)

**Fig 9. Evaluation of spatial resolution of the cost-efficient CT scanner.** (a) a single reconstructed transaxial-slice CT image of the slanted-edge plate of the CT quality assurance phantom. (b) reconstructed transaxial-slice CT image of the resolution coil plate of the CT quality assurance phantom illustrating the qualitative display of resolution coils.

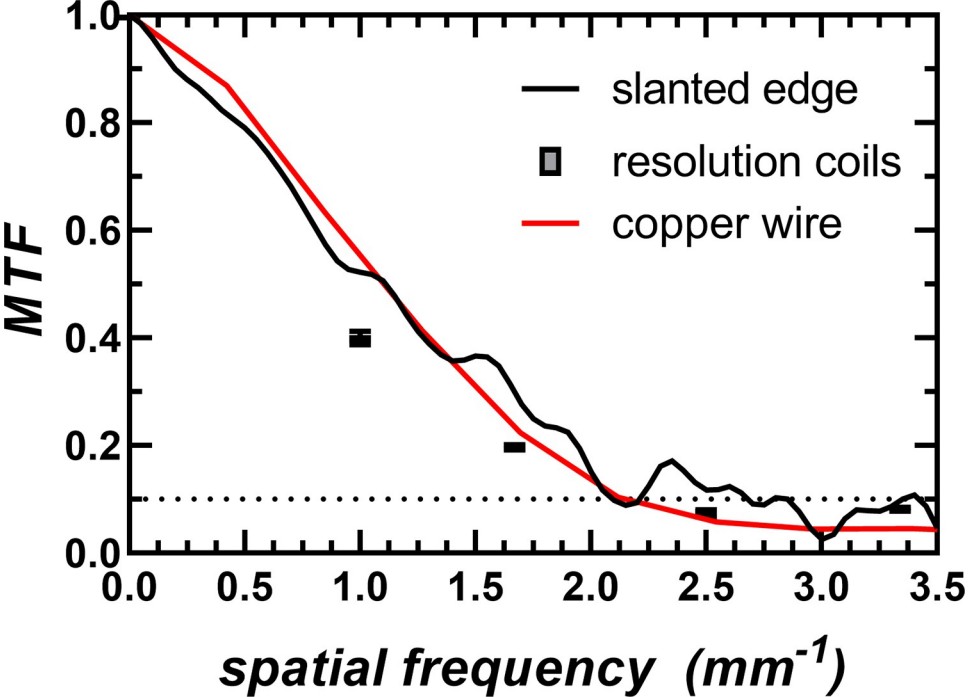

**Fig 10. Modulation transfer function (MTF) of the cost-effective CT scanner measured from the slanted-edge plate, the resolution coil plate, and the copper wire of the CT quality assurance phantom.** The 10% MTF level was reached at 2.12 mm$^{-1}$, corresponding to a spatial resolution of 235 μm.

2.12 line pairs per mm, which corresponds to a spatial resolution of 235 μm. The same results were observed and confirmed when calculating the MTF of the system using the wire-based methodology (Fig 10).

Qualitatively, it was observed that only the 300 μm and the 500 μm coil patterns were resolved, as they represented spatial frequencies of 2.5 and 3.3 line pairs per mm respectively (Fig 9B). The MTF estimates using the standard deviation of ROIs inside the coil patterns closely agreed with the other quantitative MTF calculation methods (Fig 10).

**b. Geometric accuracy.** The average, in-plane, voxel spacing of the CT reconstructions (118.39±0.20 μm) was calculated by dividing the known distance between beads in the geometric accuracy plate of the QA phantom (Fig 11) by the measured inter-bead distance in voxels. Table 3 shows the sub-voxel Euclidean distances between the centroids of all five beads of the geometric-accuracy plate of the QA phantom. The mean sub-voxel physical distances for all three known inter-bead configurations (34.98±0.06, 24.74±0.01, and 49.47±0.03 mm) were 295.63±0.46, 209.05±0.46, and 418.10±0.16 voxels, respectively. The small standard deviation between the measured Euclidian distances shows the high geometric accuracy of the CT reconstructions achieved by the low-cost CT system.

**c. Linearity.** Linearity (p < 0.001) was calculated using the measured intensity values of ROIs placed within vials filled with various concentrations of iodine (Fig 12A). Fig 13 shows the relationship between signal intensity (S) and iodine concentration (C), as well as the result of a linear regression between these variables S = 54.62 (mg ml$^{-1}$) x C + 26.4 (HU), where C is the iodine concentration in mg ml$^{-1}$ (R$^2$ = 0.9998). The y-intercept 26.4 HU was not significantly different than zero meaning that the system CT number was well calibrated.

**d. Uniformity and noise.** Quantitatively, the difference in CT numbers between the periphery of the QA phantom and the center was 1.73% (~ 18 HU). Qualitatively, the

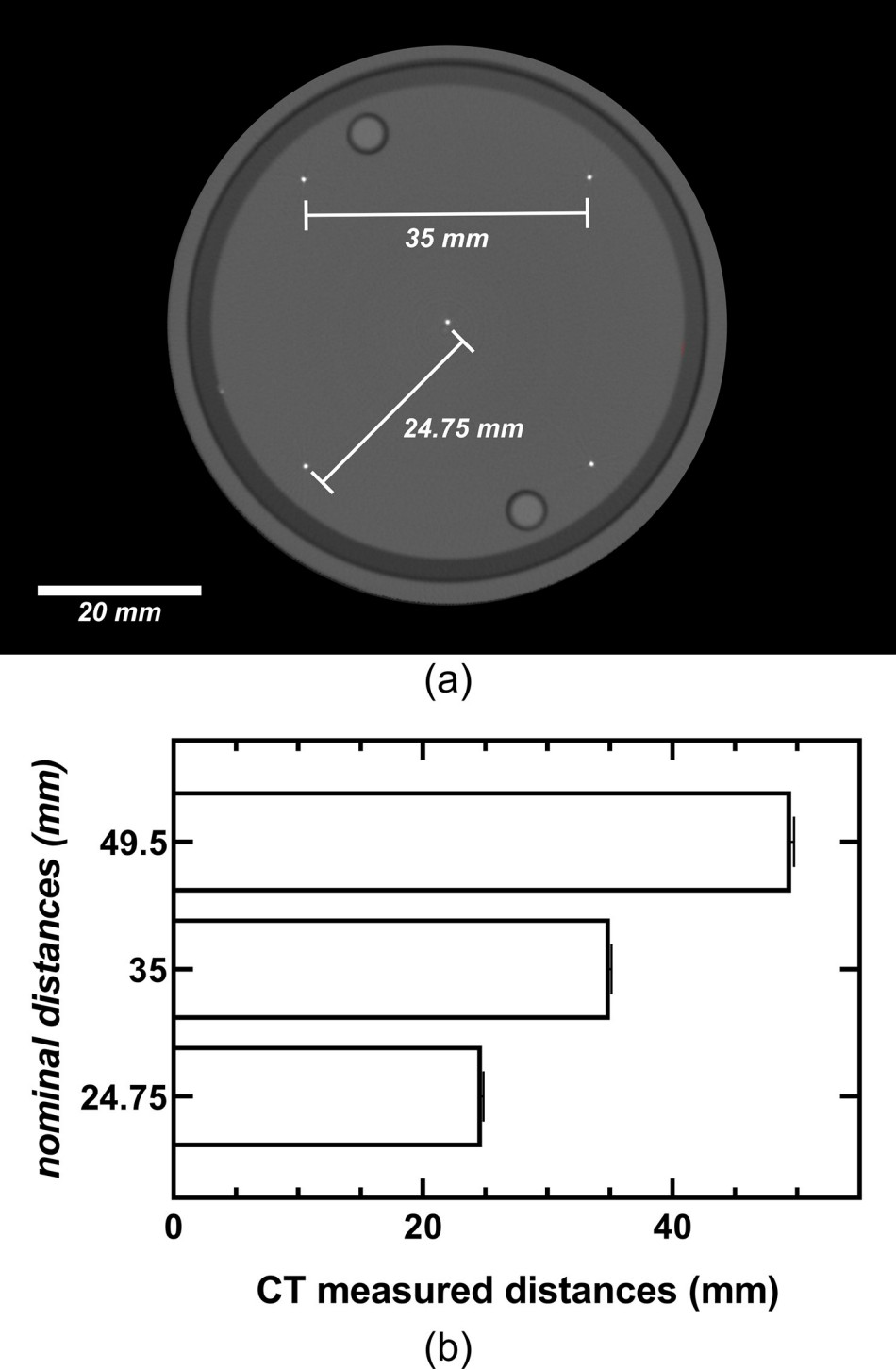

**Fig 11. Geometric-accuracy plate.** (a) reconstructed transaxial-slice CT image of the geometric-accuracy plate of the CT quality assurance phantom with four beads located at the periphery and 34.98 ±0.06 mm apart and one central bead at a distance of 24.74 ±0.01 mm from the other four. (b) all possible nominal versus CT calculated distances between beads depicting error bars that represent the 95% confidence interval.

**Table 3. Euclidean distances, in mm, between the centroids of all five beads of the geometric accuracy plate of the CT quality assurance phantom.**

| | Central bead | Periphery 1 | Periphery 2 | Periphery 3 | Periphery 4 |
|---|---|---|---|---|---|
| Central bead | 0 | 24.74 | 24.83 | 24.74 | 24.68 |
| Periphery 1 | 24.74 | 0 | 34.99 | 49.48 | 35.00 |
| Periphery 2 | 24.83 | 34.99 | 0 | 35.09 | 49.52 |
| Periphery 3 | 24.74 | 49.48 | 35.09 | 0 | 34.92 |
| Periphery 4 | 24.68 | 35.00 | 49.52 | 34.92 | 0 |

uniformity of the system was assessed with the radial signal profiles taken through the center of the uniformity plate of the QA phantom (Fig 14B). Noise (54.48±4.6 HU) was characterized using the average standard deviation of the five ROIs used for the uniformity analysis. Noise increased up to 82 HU for ROIs within materials with higher attenuation coefficients such as the highest-concentration (30 mg ml$^{-1}$) iodine solution of the linearity plate. This is a good indicative that the system is dominated by photon noise.

## C. Titanium metal-alloy (Ti6Al4V) phantoms

**a. Porous gyroid-based scaffold.** As shown in Fig 15A, the low-cost CT system produced a CT reconstruction of the 17 x 40 mm titanium-alloy porous cylinder that closely matched the prescribed CAD design. Inspection of the reconstructed slices clearly shows the gyroid-based internal structure of the scaffold allowing full interrogation of the reconstructed volume (Fig 15B, 15D–15F). Fig 15C shows how the threshold value was determined in order to segment the titanium ROI using the FWHM method. The total porosity of the construct (84.26%) was calculated by subtracting the volume of the titanium ROI (1429.95 mm$^3$) from the volume of the bounding, cylindrical ROI (9081.95 mm$^3$). The outline of the cylindrical ROI corresponding to 100% of the volume of the part is shown in Fig 15F.

The porosity of the gyroid-based titanium-alloy scaffold was also estimated using the mass of a solid cylinder (17 x 40 mm) that was printed at the same time as the scaffold. The mass of the solid cylinder was found to be 38.86 g and the mass of the porous scaffold 6.26 g, leading to a calculated porosity of:

$$porosity \% = \left(1 - \frac{scaffold\ mass}{solid\ mass}\right) \times 100 = \left(1 - \frac{6.26\ g}{38.86\ g}\right) \times 100 = 83.89\%$$

**b. Titanium-alloy resolution phantom.** Results from 3D micro-CT analysis with the high-energy commercial system (Zeiss Xradia 410 Versa) showed that the nominal dimensions of the internal defects were, on average, 170 μm less than the as-built dimensions. Specifically, the 25, 50, 100, 150, 300 μm defects were measured as 190, 220, 270, 320, and 480 μm, respectively. This discrepancy is likely related to the beam-spot compensation algorithm of the commercial 3D metal printer. The high-energy scan data also confirmed that the internal defects were fabricated consistently throughout the phantom, from the top to bottom slices. Fig 16 shows the low-cost CT reconstruction of the conical titanium-alloy resolution phantom. Fig 16B and 16D show photon starvation artifacts starting around the 13 mm diameter mark and worsening at greater diameters. This demonstrated that some medical components would need to be carefully loaded into the scanner's rotational stage to ensure that x-ray paths travel across the object with the smallest possible path-length across metal. Below this level, most of the internal features of the phantom were still detected, with the exception of the cylindrical voids with CT measured diameters smaller than 0.25 mm, and the nominal 10 lp/mm bar

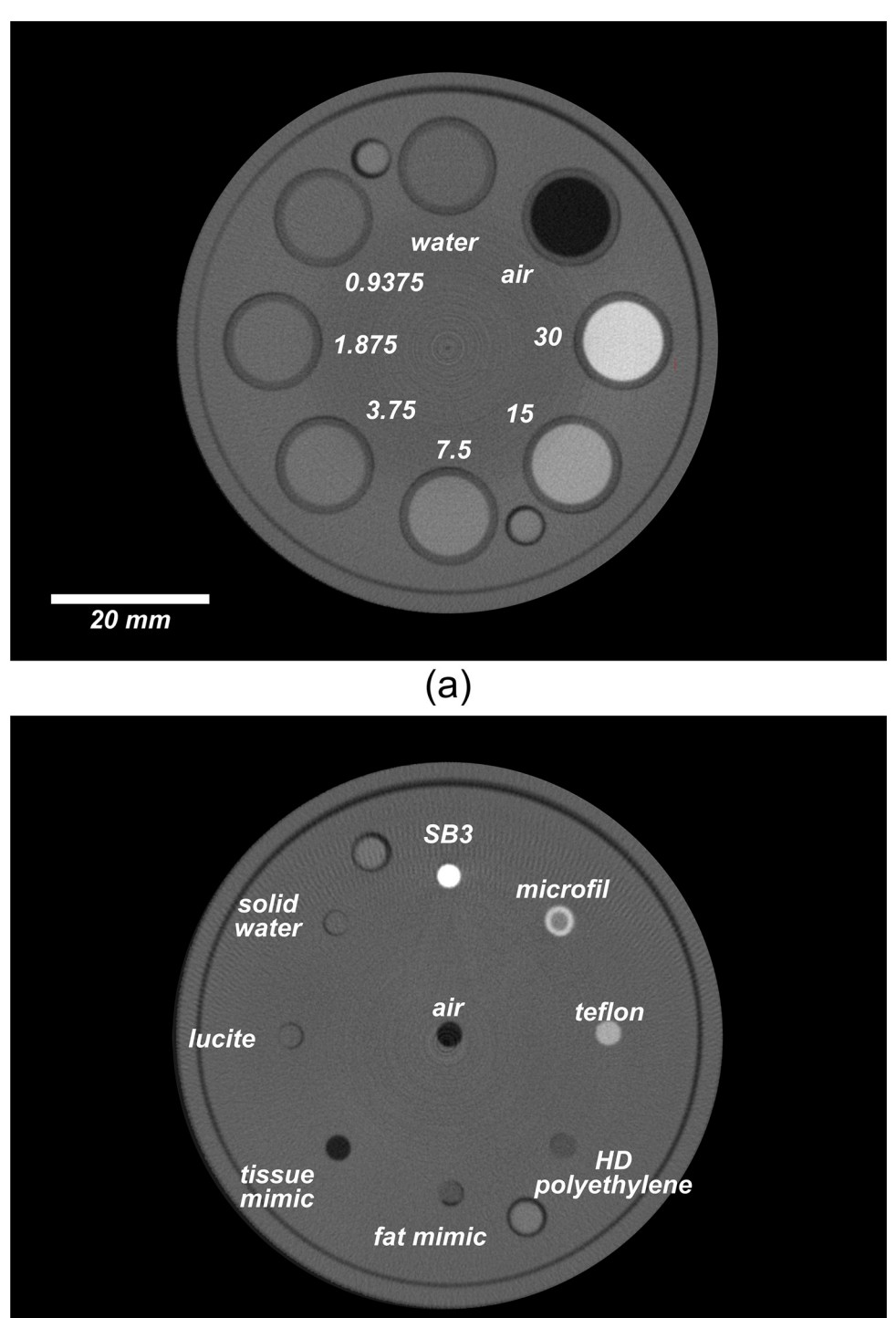

**Fig 12.** Reconstructed transaxial-slice CT images of the linearity plate (a) with air, water, and various concentrations of iodine shown in mg ml$^{-1}$; and the CT number evaluation plate (b) of the CT quality assurance phantom.

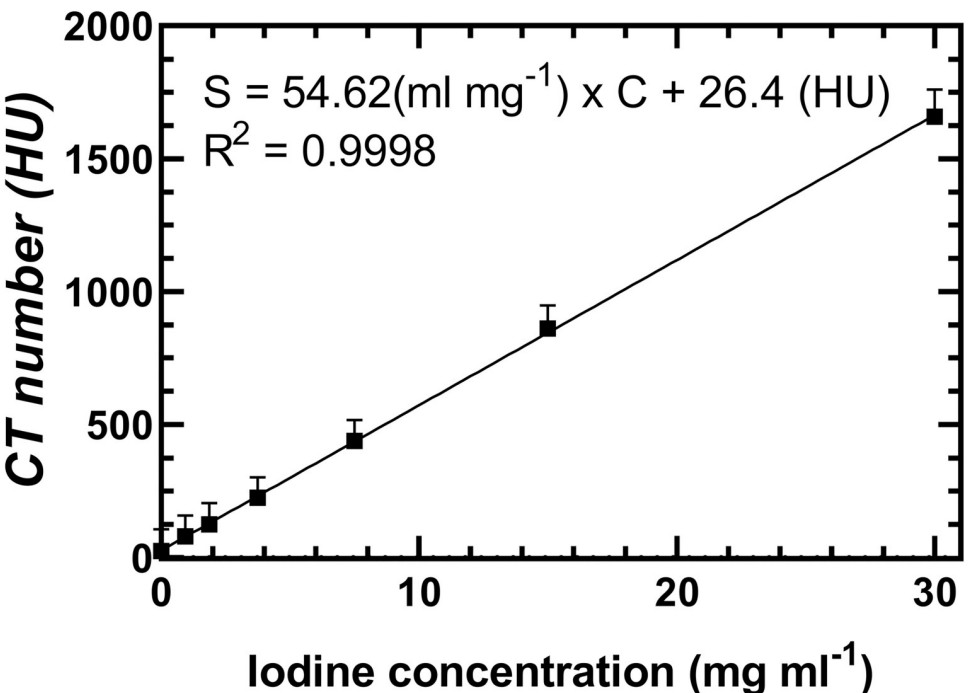

**Fig 13. Plot of measured CT number within ROIs placed in each iodine vial versus known iodine concentration within the linearity plate of the CT quality assurance phantom.** A significant linear correlation is seen between CT number in HU and iodine concentration in mg ml$^{-1}$.

patterns. Fig 16C and 16E show that defects larger than 0.5 mm would be characterizable using the proposed CT system. Fig 16F demonstrates that as the diameter of metal increases, noise levels in the reconstructed image increase, as well reducing the conspicuity of the internal voids.

## IV. Discussion

Non-destructive testing of porous titanium-alloy 3D-printed parts is possible using a low-cost micro-CT scanner fabricated with off-the-shelf components. The cost-effectiveness of the proposed system takes advantage of some of the design constrains characteristic of most medical devices–which include high-porosity, small size, and relatively-low x-ray attenuating materials. This strategy demonstrated that routine NDT of titanium 3D-printed medium-sized highly-porous medical components is feasible using the proposed cost-effective CT system.

This novel CT system includes a low-cost lens-coupled x-ray detector comprised of a consumer-grade dSLR camera and a phosphor screen. The consumer grade dSLR camera and an appropriately chosen lens exploit the front-lit configuration described by Panna et al. [18], where the light emitted by the phosphor is more effectively captured than in other configurations [13, 14, 17, 18, 28]. Other advantages of this configuration include: (1) a reduced optical diffusion of the captured light as it does not need to pass through the phosphor to be detected (i.e., reduced light scattering within the phosphor); (2) improved spatial resolution in the trans-axial plane due to the tilted configuration of the phosphor screen; (3) no bright-pixels in the captured images as the camera sensor can be positioned out of the direct x-ray beam path and properly shielded; and (4) a significant reduction in cost compared to similar-sized, digital, flat-panel detectors.

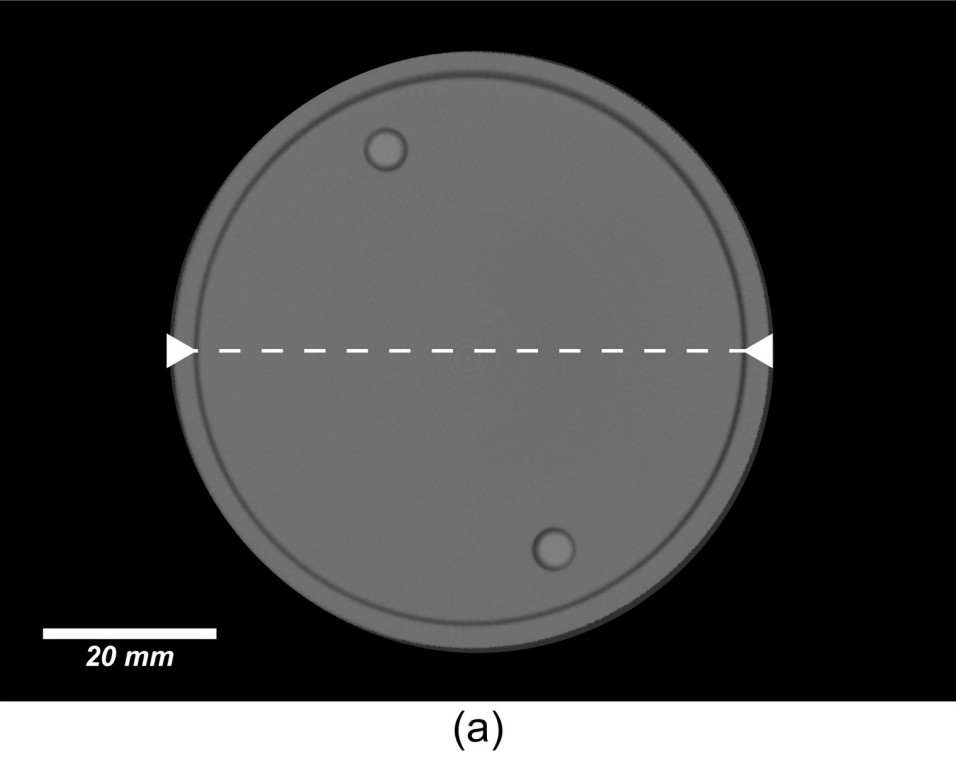

(a)

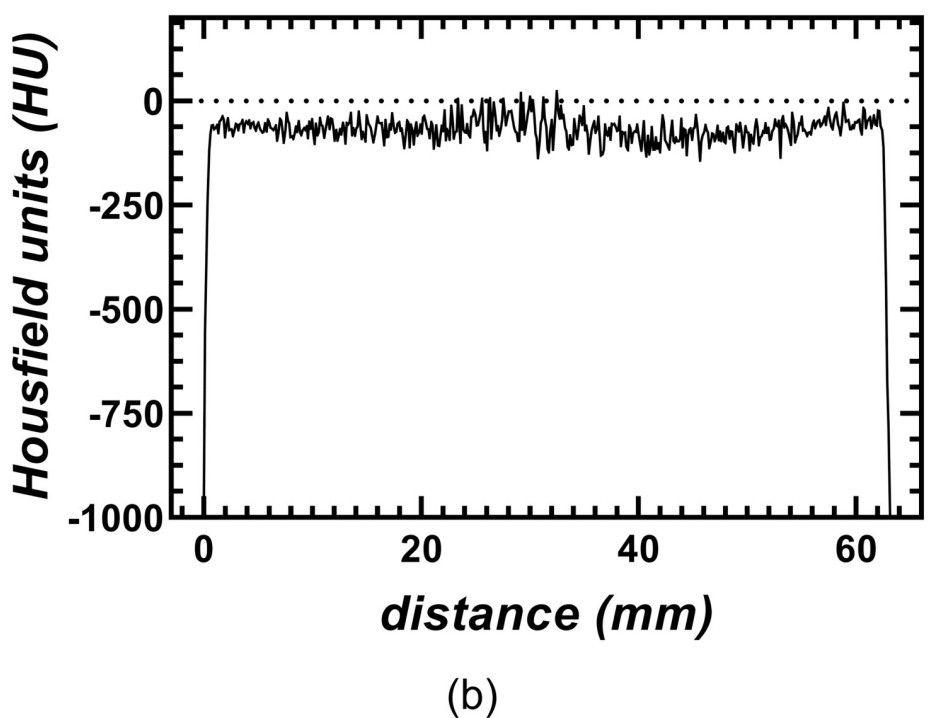

(b)

**Fig 14.** (a) Reconstructed transaxial-slice CT image of the uniformity plate of the CT quality assurance phantom. (b) Radial signal profile taken through the center of the uniformity plate as illustrated by the white dotted-line in (a).

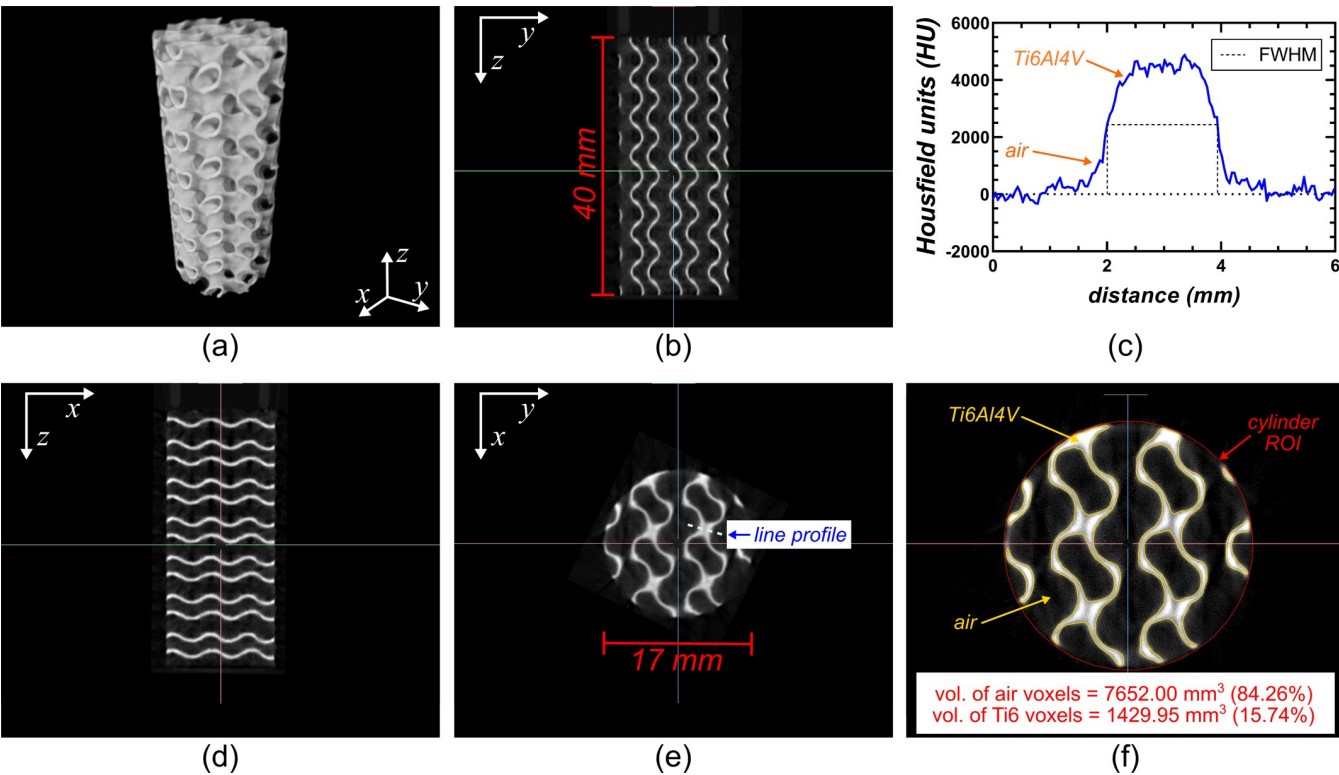

**Fig 15. CT reconstruction of the porous gyroid-based cylindrical titanium-alloy (Ti6Al4V) scaffold using the cost-effective CT scanner.** (a) perspective view of volumetric rendering using the FWHM threshold used to segment titanium. (b) Trans-coronal slice CT reconstruction of the porous cylinder. (c) Signal profile across a wall of the porous cylinder used to determine the FWHM threshold for segmentation. (d) Trans-sagittal slice CT reconstruction of the porous cylinder. (e) Transaxial slice CT reconstruction of the porous cylinder and line profile used for (c). (f) close-up version of (e) illustrating boundaries between the titanium and air ROIs used to measure the porosity of the porous, cylindrical, titanium scaffold.

For this study, effective spatial resolution (0.235 mm) was mainly limited by penumbral blur due to the focal-spot size (~ 0.3 mm), this could be easily resolved by using an x-ray source with a smaller focal-spot. For instance, if an x-ray source like the proposed Sourceblock SR-80-500 were to be used, then the focal-spot blur will be ten times lower. This will take better advantage of the high resolution of the camera, lens, and the phosphor screen. Further improvements in spatial resolution could be achieved by removing the one red, two green, and one blue (RGGB) Bayer filter included in the Nikon D800 dSLR camera, as in this implementation only the green channels used the full 14-bit dynamic range of the camera's detector. This modification could also improve the light sensitivity of the detector, as the added filters reduce the quantum efficiency of the underlying pixels. The selection of a color-camera was based on cost and availability considerations, as similar true black and white cameras are more expensive and difficult to source.

Although the projection images in this study captured a field-of-view covering the full area of the (180 by 240 mm) phosphor screen, both the detector-to-source distance and the object-to-detector distance could be adapted for smaller objects. This will increase the intensity of the captured light requiring lower ISO sensitivity values to cover the full dynamic range of the CMOS detector. Lower ISO values are desirable as signal-to-noise will be reduced by half for each ISO speed setting or "f-stop". Furthermore, the camera could be set to capture smaller regions of the CMOS detector. For example, the Nikon D800 has a 24 by 16 mm DX mode, which could reduce read-out and post-processing times.

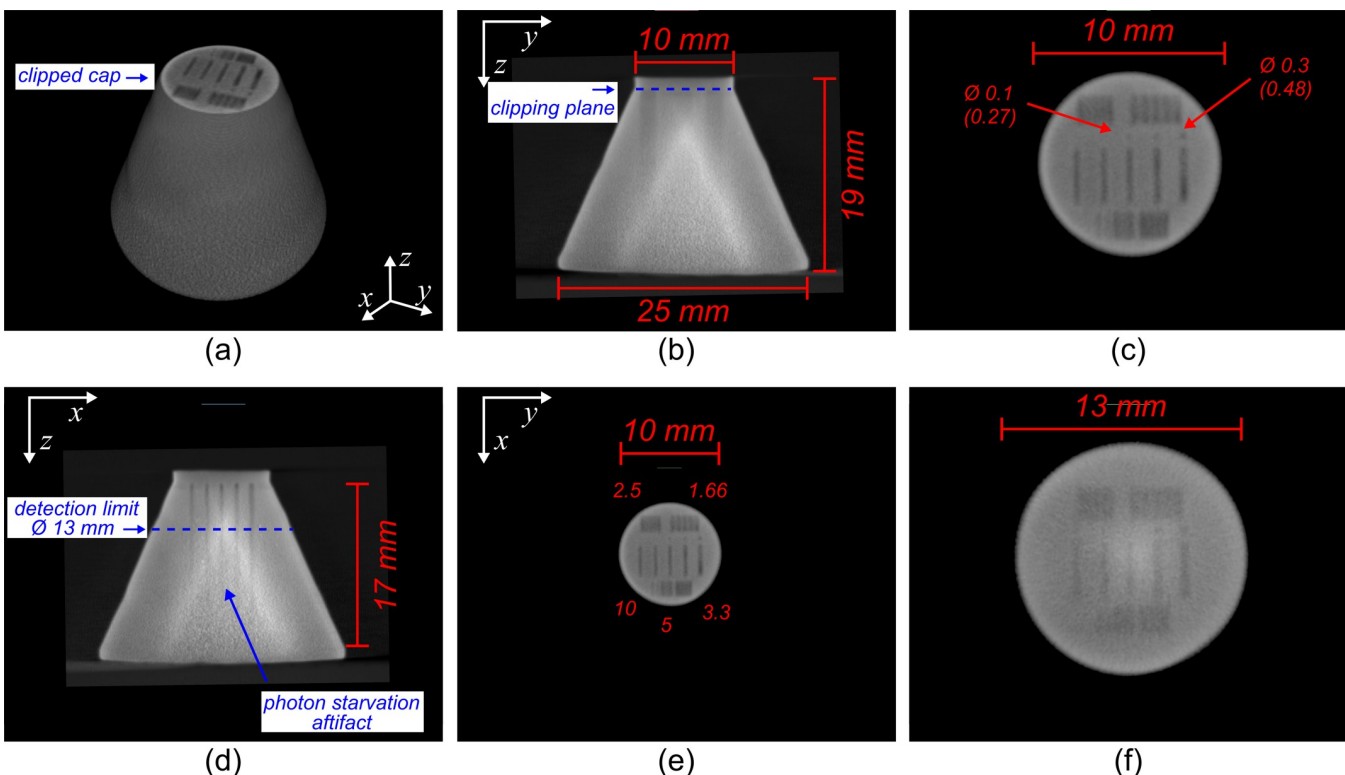

**Fig 16. Beam-hardening corrected CT reconstruction of the titanium-alloy resolution phantom using the cost-effective CT scanner.** (a) Volumetric rendering of the phantom with the top cap clipped to reveal the internal-void features of the resolution phantom. (b) Transcoronal slice of the resolution phantom showing the relative position of the clipping plane for (a), (c) and (e). (c) Close-up of (e) illustrating the internal features of the phantom at the level of the clipping plane where the outer diameter of the object is 10 mm. Note that the CT-measured true dimensions are labelled in parenthesis after the nominal dimensions of the internal voids. (d) Transsagittal slice of the resolution phantom showing the level where the resolution limit was reached. (e) Transaxial slice of the resolution phantom showing details of the internal features of the phantom at the level of the clipping plane labelled in (b). (f) Transaxial slice at the level of the resolution limit where the external diameter of the phantom was 13 mm. At diameters greater than this, internal-void features are difficult to distinguish and photon starvation artifacts start to dominate.

Changes in the geometry of the system, to accommodate various specimen sizes, or following assembly of the system, will require the acquisition of new geometric calibration images. This process was not time-consuming, except for the manual segmentation of the markers of the calibration grid. A more sophisticated and automated segmentation algorithm would be recommended in case of regular modifications to the geometry of the system. Another alternative would be to acquire calibration images for a set of pre-calibrated geometries using additional repositioning hardware. The geometric accuracy of the reconstructions in this study shows that the proposed geometric calibration strategy was successful. Future work includes the manufacturing of a calibration grid with a larger coverage of the phosphor screen. In such case, geometric calibration might be more challenging at the edges of the full-frame image, as non-linear lens aberrations are more pronounced in this area of the image–especially when using large lens aperture settings [16].

Scan time is another important aspect to consider for the cost-effectiveness of routine NDT. Time-consuming NDT might cause prohibitively long delays in some clinical applications or significantly increase the cost per scan [5]. In this study, the total scan time per projection image was 7.5 seconds (i.e., 45 minutes for 360 projection images). From these 7.5 seconds, 4 s were utilized for data acquisition, and 3.5 s were required for data transfer from the Nikon camera to the computer over USB 3.0. The reconstructions achieved by the

proposed low-cost scanner suggest that acceptable image quality can be achieved despite the number of projections being lower than the suggested by the strict Nyquist criterion. A reduced number of projections has been suggested in the literature as one of the most effective strategies to reduce scan time, while maintaining image quality [5].

The proposed low-cost micro-CT scanner closely matched the geometric accuracy, noise, uniformity, and linearity of commercially available micro-CT scanners [19]. Furthermore, the proposed system may be able to outperform scanners that use fiber-optic tapered detector technologies, as these are often more susceptible to CT ring artifacts. Ring artifacts are caused by non-linear response of the x-ray detector elements (i.e., fixed-pattern noise), which can be caused by severe defects of the detector itself, or the readout electronics [29]. In the implemented lens-coupled detector system, fixed-pattern noise is more likely to arise from discontinuities or defects in the surface of the phosphor screen and less likely to be present at the level of the camera's sensor. This is due to the high degree of quality control in the commercial digital-photography industry, where detectors with many critical defects are unlikely to be sold.

Additive manufacturing has gained popularity in the medical and dental sectors thanks to its ability to produce highly-porous, but strong, medical components. The proposed system has demonstrated the capability to interrogate titanium-alloy objects up to a total x-ray path length through titanium of 13 mm. This was shown by the accurate volumetric representation of a 17 mm diameter gyroid-based 80% porous scaffold. The characterization of internal-voids in the titanium resolution phantom matched the expected performance with a voxel resolution of 118 μm. Fig 16 shows the clear identification of all defects above 270 μm. These detected defects were smaller than the suggested 0.5 mm size for mechanically critical cracks or voids, that–if unnoticed–could seriously compromise the integrity of the 3D printed part. Note that 3D printed parts can also be susceptible to cross-contamination with other metals or metal oxides during fabrication. The proposed system was limited to the identification of internal defects only and it is not anticipated to aid in the identification of such cross-contaminants.

Although the main objective of this study was to demonstrate that the proposed system could be used for routine NDT of titanium-alloy 3D-printed parts its flexibility, portability, and versatility can be exploited in other applications. For example, the system could be used for volumetric imaging of archaeological samples in remote sites. In such cases, the included, full-frame, dSLR camera could also be used independently for sample documentation or 3D photogrammetry. Space exploration missions could also exploit this proposed strategy, which, again, will allow the included camera to be employed in a variety of tasks.

The low-cost scanner in this study was safely operated inside an adequately shielded room. If this setup is not feasible, the frame of the scanner will need to be fully covered using proper shielding (e.g., lead panels). The additional cost of the required shielding will not significantly impact the final cost of the scanner. However, if the scanner were to be used in remote sites, the required shielding might interfere with the transportation of the equipment and therefore the scanner might not be safe to operate under normal conditions. A solution for this scenario will be to define a designated controlled area where no personnel might be present. Similar strategies are often implemented in industrial, temporary sites or large-animal, veterinary radiography. Wireless-interlock technology could be used to define a perimeter at the boundary of the designated controlled area to avoid accidental exposure to ionizing radiation.

Another advantage of the proposed system is that it can be easily updated. For instance, the system's camera could be periodically replaced as digital photography technology continues to evolve. Newer technologies could improve spatial resolution, reduce data transferring time, add additional capabilities to the system, or increase the lifespan of the image acquisition hardware. For example, mirrorless cameras do not require shutter replacement or maintenance, which may be needed for the Nikon D800 after 200,000 actuations. Additionally, the use of a

mirrorless camera could enhance the fluoroscopy capabilities of the proposed system, as these cameras may be better suited for video streaming. Finally, as demonstrated by the use of a general-purpose, x-ray source, the described hardware, data-acquisition techniques, post-processing, and data reconstruction are compatible with most commercially-available x-ray units, which adds another layer of flexibility to the low-cost scanner.

From a cost-effectiveness standpoint, the main components of the described scanner can be purchased at a fraction of the cost of commercially-available, micro-CT scanners with comparable imaging characteristics. Even considering that in the industry the cost of the components might only represent a third of the cost of the device ($ 85000 for a $ 260000 scanner) the costs of the described system (< $ 11000) are still significantly lower. Using the proposed CT scanner, the cost-per-scan will be significantly lower and will provide a cost-effective solution for non-destructive testing of medium-sized titanium-alloy medical components. Furthermore, the possibility of using the included dSLR camera for other quality assurance tasks could further warrant the cost-effectiveness of the equipment. For example, the camera could be refocused to image the part directly and be used for photogrammetry, surface inspection, and geometric evaluation. Finally, the modular nature of the design will allow updates, upgrades, and replacement of individual components of the scanner without the need to replace the entire unit.

## V. Conclusion

This study presents the successful design, fabrication, and implementation of a low-energy (80 kVp) micro-CT scanner for cost-effective routine non-destructive testing of porous 3D-printed titanium-alloy medical components. Cost-effective CT reconstructions of titanium 3D-printed samples were achieved by acquiring data at lower x-ray potential and using a low-cost lens-coupled detector system. These reconstructions closely matched the geometric accuracy, uniformity, and linearity of relatively-expensive commercial micro-CT scanners.

The low-cost detector system, utilized a phosphor screen lens-coupled to a commercially available consumer-grade full-frame dSLR camera using a front-lit tilted configuration. This configuration provided the light-collection efficiency required for CT imaging. The camera's sensor size (36 by 24 mm), the sensor technology (CMOS), and the lens light-transmission (f/1.4 fast lens) were key factors to ensure that the noise characteristics of the detector were dominated by photon counting statistics, rather than electronic or quantum noise.

Geometric distortions in the projection images acquired using the tilted-detector lens-coupled phosphor screen were successfully corrected using a Cartesian calibration grid. The effectiveness of the correction was evaluated at the level of the projection images and by assessing the performance of the scanner using a comprehensive CT quality assurance phantom. The performance evaluation of the system showed quantitatively accurate and geometrically stable reconstructions, with acceptable noise levels and image quality.

The proposed design for the cost-effective scanner allowed non-destructive evaluation of medium-sized titanium-alloy 3D-printed parts with a thickness up to 13 mm of solid metal. Since this evaluation was done using test-objects, future work includes non-destructive testing of medical components with clinically relevant geometries. It can be anticipated that components with large amounts of solid metal will require a modified imaging protocol to overcome the penetration limits of the 80 kVp x-ray source.

## Acknowledgments

D.W.H. is the Dr. Sandy Kirkley Chair in Musculoskeletal Research within the Schulich School of Medicine & Dentistry at Western University. S.F.C. is supported in part by a

Transdisciplinary Bone & Joint Training Award from the Collaborative Training Program in Musculoskeletal Health Research at The University of Western Ontario. The authors would also like to thank the technical staff at ADEISS for their assistance with the manufacturing of the titanium, 3D printed, phantoms used for this study, and Hristo Nikolov for his technical support conditioning the CT quality assurance phantom prior data acquisition and manufacturing of the scanner's frame.

## Author Contributions

**Conceptualization:** David Wayne Holdsworth.

**Formal analysis:** Santiago Fabian Cobos.

**Funding acquisition:** David Wayne Holdsworth.

**Investigation:** Santiago Fabian Cobos.

**Methodology:** Santiago Fabian Cobos, David Wayne Holdsworth.

**Project administration:** Santiago Fabian Cobos, David Wayne Holdsworth.

**Resources:** David Wayne Holdsworth.

**Software:** Santiago Fabian Cobos, Steven Ingo Pollmann.

**Supervision:** David Wayne Holdsworth.

**Validation:** Santiago Fabian Cobos, David Wayne Holdsworth.

**Visualization:** Santiago Fabian Cobos.

**Writing – original draft:** Santiago Fabian Cobos, Christopher James Norley.

**Writing – review & editing:** Santiago Fabian Cobos, Christopher James Norley, Steven Ingo Pollmann, David Wayne Holdsworth.

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
