## [Decision Letter · Decision Letter 0]

1 Apr 2022

PONE-D-21-33035Cost-effective micro-CT system for non-destructive testing of titanium 3D printed medical componentsPLOS ONE

Dear Dr. Cobos,

Thank you for submitting your manuscript to PLOS ONE. After careful consideration, we feel that it has merit but does not fully meet PLOS ONE’s publication criteria as it currently stands. Therefore, we invite you to submit a revised version of the manuscript that addresses the points raised during the review process.

We look forward to receiving your revised manuscript.

Kind regards,

Alessandra Giuliani

Academic Editor

PLOS ONE

Journal Requirements:

Additional Editor Comments (if provided):

This paper presents a challanging idea to make X-ray CT more accessible; however there are several points which require significant revision to make your study publishable.

Reviewers' comments:

Reviewer's Responses to Questions

**Comments to the Author**

1. Is the manuscript technically sound, and do the data support the conclusions?

Reviewer #1: Yes

Reviewer #2: Yes

2. Has the statistical analysis been performed appropriately and rigorously? 

Reviewer #1: Yes

Reviewer #2: No

3. Have the authors made all data underlying the findings in their manuscript fully available?

Reviewer #1: Yes

Reviewer #2: Yes

4. Is the manuscript presented in an intelligible fashion and written in standard English?

Reviewer #1: Yes

Reviewer #2: No

5. Review Comments to the Author

Reviewer #1: This is an interesting experimental study, describing the design, manufacturing and implementation of a cost-effective micro-CT system for non-destructive testing of 3D printing metal parts, including titanium biomedical components fabricated using laser powder-bed-fusion technology. The scanner system allows to survey the porosity and cracks in titanium parts with thicknesses of up to 13 mm of solid metal. Quantitatively, the scanner produced geometrically-stable reconstructions, with a voxel size of 118 μm, and noise levels under 55 HU. In addition, scanner could be introduced as part of the process control and validation of component quality.

The paper is well written. There are a few minor revisions that should be made for it:

1) various concentrations of iodine shown in ml mg-1 (line 364)

The unit of measure ml mg-1 should be corrected.

2) Furthermore, thanks to the relatively high difference between the CT number of air and titanium, internal-void features or defects were identifiable below the spatial resolution limits of the system. (line 485-487)

The description of additive manufacturing defects detectable is limited to void defects. It is know that AM technologies, especially LPBF, deals with cross-contamination issues. Is the system able to detect also eventual cross-contamination (other metals or material oxide)? It is recommended to implement the argumentation on AM defects detectable.

3) Although these detected defects were not geometrically accurate, in practice, they show that the system could be effective at flagging internal defects, such as cracks or voids, that – if unnoticed – could seriously compromise the mechanical integrity of the 3D printed part. (line 488-491)

The flagging of internal defect below the instrument resolution provides only a qualitative indication of production process suitability. To evaluate the mechanical integrity of sample, it is necessary to quantify the porosity on effective area, for avoiding to not accept suitable samples (ex. for static application, a certain level of porosity are acceptable due to ASTM specifications). Please comment in the revised version of the manuscript.

4) We believe the cost-per-scan will be significantly lower when using our proposed CT scanner design, and will provide a cost-effective solution for non-destructive testing of medium-sized, titanium-alloy, medical components. (line 522-524)

Please explain in detail which are the “medium-sized, titanium-alloy, medical components” considered as target of this work.

Reviewer #2: please see the attached

This paper presents a fantastic idea to make X-ray CT more accessible in the classroom and a number of other situations. However there are a number of problems which require significant revision to make this publishable. I hope the authors do take the time to make the proposed edits as I believe it to be useful to others in the community. A few core issues

- Some incorrect sweeping statements have been made. I have highlighted these where I can

- Some of the calibration parts could be better described.

- There are objects which contain dimensional measurements and are used to quantify the ability of the system. There is no evidence that these have been calibrated or measured. This needs looking at and have suggested below

- Please, please, please identify which commas are actually necessary

6. PLOS authors have the option to publish the peer review history of their article (what does this mean?). If published, this will include your full peer review and any attached files.

Reviewer #1: No

Reviewer #2: No

---

## [Author Response · Author response to Decision Letter 0]

14 Jul 2022

Letter of response to reviewers

Manuscript titled:

Cost-effective micro-CT system for non-destructive testing of titanium 3D printed medical components

Response to reviewer #1

Reviewer’s comment:

1) various concentrations of iodine shown in ml mg-1 (line 364)

The unit of measure ml mg-1 should be corrected.

Response:

First, we want to thank the reviewer for taking the time to carefully review our manuscript. We appreciate all the observations and believe that the applied modifications have significantly improved our text. We apologize for the oversight of this typographic error. We have corrected the units to mg ml-1.

Reviewer’s comment:

2) Furthermore, thanks to the relatively high difference between the CT number of air and titanium, internal-void features or defects were identifiable below the spatial resolution limits of the system. (line 485-487)

The description of additive manufacturing defects detectable is limited to void defects. It is known that AM technologies, especially LPBF, deals with cross-contamination issues. Is the system able to detect also eventual cross-contamination (other metals or material oxide)? It is recommended to implement the argumentation on AM defects detectable.

Response:

We thank the reviewer for bringing the potential of cross-contamination of the 3D printed part during manufacturing. We do not anticipate that our system will be able to aid in the identification of these contaminants. We have included clarification about the scope of the NDT in the discussion section of the manuscript.

Reviewer’s comment:

3) Although these detected defects were not geometrically accurate, in practice, they show that the system could be effective at flagging internal defects, such as cracks or voids, that - if unnoticed - could seriously compromise the mechanical integrity of the 3D printed part. (line 488-491)

The flagging of internal defect below the instrument resolution provides only a qualitative indication of production process suitability. To evaluate the mechanical integrity of sample, it is necessary to quantify the porosity on effective area, for avoiding to not accept suitable samples (ex. for static application, a certain level of porosity are acceptable due to ASTM specifications). Please comment in the revised version of the manuscript.

Response:

We thank the reviewer for identifying this topic and for his suggestion to be more descriptive with regards to the specific characteristics that cracks or voids should have in order to be considered critical defects in the part. We have included the following text that is meant to clarify these aspects of the NDT evaluation.

“It should be emphasized that routine non-destructive testing of medical components should not have the objective of producing volumetric data for high-precision metrology of the parts but rather used for an assessment of critical defects with the potential of affecting the mechanical properties of the component. This assessment should include the size, location, and distribution of the defects. In the case of titanium medical components, the critical defect size might be 0.5 mm, requiring a spatial resolution for the CT volume in the order of 0.2 mm in order to visualize any mechanically critical defects.”

Reviewer’s comment:

4) We believe the cost-per-scan will be significantly lower when using our proposed CT scanner design, and will provide a cost-effective solution for non-destructive testing of medium-sized, titanium-alloy, medical components. (line 522-524)

Please explain in detail which are the "medium-sized, titanium-alloy, medical components" considered as target of this work.

Response:

We thank the reviewer for pointing out the need to provide specific examples to illustrate the objects that could benefit of being tested using our low-cost scanner. We have included a list of a medical devices that can be manufactured using 3D printing and that will be suitable for being evaluated using our NDT system. (Last paragraph of introduction)

 

Response to Reviewer #2

Reviewer’s comment:

“fluence” has been used instead of fluorescence throughout. Can you change assuming this was the intent. Its not a term I’ve come across in my time in tomography, and was unable to find a definition. Further many of the papers you cite us the term fluorescence so is the expected terminology of their readership.

Response:

First, we want to thank the reviewer for all their insightful comments we have enjoyed taking a closer look to this manuscript, and believe that all the required changes have made it an improved document.

We acknowledge that the use of the term fluence (defined as the quotient dN by dA, where dN is the number of x-ray photons that enter an imaginary sphere of cross-sectional area dA) could generate some confusion among readers. We propose to improve the text by using the term “flux” instead. Photon flux is defined as the number of photons per second per unit area, and is the more appropriate term for the context of our paper. Where we use the term to refer to an increase in photon count across a unit area per unit time. We are sorry that the use of the term “fluence” lead to confusion to the unrelated term fluorescence.

Reviewer’s comments:

There are far too many commas throughout the document to the point of being irritating. Please remove them as 75% of them are not required.

Line 29 “the cost-effective….”

This sentence has a horrendous amount of commas. Please sort.

Line 119 This sentence is a mess. Its too long and needs splitting and has far too many commas. The first three commas are the unnecessary ones

Line 127 the two commas are unnecessary

Even the first sentence has a ridiculous amount of commas!

Again the horrendous amount commas in this first sentence is infuriating.

Response:

We are sorry that the writing style of our manuscript has made it not enjoyable to read. We acknowledge that the repetitive use of several adjectives before a noun has caused distraction for the reader and prevented an efficient lecture of our text. We have noticed that writing styles treat this grammatical structure differently and have decided to adhere to the Oxford Style Guide, which does not require the use commas between multiple classifying adjectives. We have implemented this modification across the whole document.

Reviewer’s comments:

Often you slip into first person with “we” and “our”. Please remove

Line 30 “We believe…”

Make third person. First person not appropriate style, and correct throughout abstract

Line 78 – remove all first person and make third person

Line 293 “thanks to the use” language is too casual for a scientific paper. In fact I think a lot of section A has this sort of tone which could be improved.

Line 427 “we” remove all first person. Make third person

Line 461 “in our experience” – not first person please! Change to third!

Line 531, 533 “we” remove,, line 537 “our”, make third person

Response:

Again, we are sorry that our use of language has caused distraction and portrayed our descriptions as casual or less professional. We have modified all instances of first-person sentences across the document and removed casual language.

Reviewer’s comment:

Line 24 “The imaging performance of the system is characterized using a state-of-the-art, CT, quality-assurance phantom, and two titanium, 3D-printed, test specimens”

I didn’t like this sentence. “state of the art” is not true, they are printable and accessible by anyone. Instead could you replace with a sentence that describes the key features they test. E.g. The system was tested with a number of phantom containing X to evaluate Y. Arguably this sentence isn’t even necessary

Response:

We apologize for the misleading use of the term “state-of-the-art”. It was not our intention to claim that the CT quality assurance phantom used in this study is printable or easily accessible. We have decided that the term “comprehensive” represents the characteristics of this device more accurately. 

Reviewer’s comment:

I think there are a couple of useful review papers which can provide further context on how the technology of X-ray CT has developed over time. It can then be commented that while the technology has developed, cost has been forgone to maintain the greatest scientific potential. Reviews for this discussion would be:

- P J Withers et al (2021) X-ray Computed Tomography, Nature Reviews Method Primers 1(18)

- S R Stock (2008) Recent Advances in X-ray microtomography applied to Materials, International Materials Reviews 53(3)

- S R Stock (1999) X-ray microtomography of Materials, International Materials Reviews 44(4)

Another relevant review to include, specific because you mention additive manufacture

- A Thompson et al (2016) X-ray Computed Tomography for additive manufacturing: a review, Measurement Science and Technology

Maybe important because it talks about parameter selection, but does focus on the “speed” aspect

- E A Zwanenburg et al (2022) Review of high-speed imaging with lab-based x-ray computed tomography, Measurement Science and Technology

These would help strengthen the discussion. Also Withers and Zwanenburg would benefit your paper being newer publications as a number of yours are older e.g. your first couple are 2014, 2017, 2016, 2016.

Response:

We thank the reviewer for their suggestion for additional literature to be included in the manuscript. We have included several references to these papers across the document, mainly in the introduction and discussion sections.

Reviewer’s comment:

Line 64 – I strongly disagree with your statement “in the case of medical components… spatial resolution needs to be on the order of 0.1mm”. It is very much dependent on the object and the context of what you wanted to observe! Glucose sensors for example which I have looked at needed 0.02mm. Acetabular hip cups achieved 0.03mm as per ref below, but it certainly would benefit from 0.01mm or better!

Kourra, Nadia, et al. "Computed tomography metrological examination of additive manufactured acetabular hip prosthesis cups." Additive Manufacturing 22 (2018): 146-152.

Response:

We are sorry that our text led to confusion with regards to the objectives of our proposed system. We understand that metrological CT examinations of medical devices benefit from low spatial resolutions. That being said, the intended scope of our study is to implement routine CT evaluation of medical components to assess critical defects that could compromise the mechanical integrity of the part. We have decided to include the following sentences to clarify the scope of our proposed application:

It should be emphasized that routine non-destructive testing of medical components should not have the objective of producing volumetric data for high-precision metrology of the parts but rather used for an assessment of critical defects with the potential of affecting the mechanical properties of the component. In the case of titanium medical components, the critical defect size might be 0.5 mm, requiring a spatial resolution for the CT volume in the order of 0.2 mm in order to visualize any mechanically critical defects.

Furthermore, we have included the following reference which discusses the topic of critical pore size in section 11.

Du Plessis, A., Yadroitsava, I. and Yadroitsev, I., 2020. Effects of defects on mechanical properties in metal additive manufacturing: A review focusing on X-ray tomography insights. Materials & Design, 187, p.108385.

For example, in the following paragraph the authors discuss the implications of increasing resolution in order to make CT evaluation more cost-effective:

“The fact that only roughly 0.5 mm pores and larger necessarily need to be detected can greatly simplify the X-ray tomography process and hence reduce scan times and costs. For example, scanning a 20 mm-diameter part at 20 μm voxel size allows accurate quantification of all pores >27 voxels in extent (a cube of 3x3x3 voxels), which relates to pores >60 μm. By scanning at the same quality (voltage and exposure time, etc.) but at voxel size of 100 μm, all pores >0.3 mm will be quantified but the number of projection images is reduced due to the lower magnification. This relates to scan time reduction in this example from 1 h to 10 min, for example.”

Reviewer’s comment:

Throughout section A and B you use cm, can you change all to mm. It is standard in design to use mm or m and it’s a bit odd to have cm.

Response:

We thank the reviewer for this observation we agree that keeping one unit of measurement across the document is more consistent and clearer. We have changed all measurements in cm to mm.

Reviewer’s comment:

Line 88 remove “As previously mentioned”

Line 93 too much detail. Remove “the panels were secured…”

Line 95 similarly remove “this fabric was secured…”

Line 102 similarly, reword to “The camera-lens assembly was mounted to the frame with a sliding gantry plate that allowed camera-to-screen adjustments.”

Line 104 remove reference to clips. It’s a level of detail unnecessary

Response:

We appreciate the reviewer’s suggestions to improve readability and style. We have adopted the suggested changes.

Reviewer’s comment:

What hasn’t been made clear at any point is any shielding except for that covering the camera? Did you decide not to use any? Why? Please discuss this

Response:

We thank the reviewer for bringing notice to this important point. The low-cost scanner was placed inside an x-ray shielded room and operated from a safe area. We have included these details in the methods section of the manuscript, and also included a section in the discussion describing the implications of requiring the device to be fully shielded if an x-ray lead-lined room is not available.

Reviewer’s comment:

Line 128 “to match the exposure levels”… incorporate sentence on line 130 “note that…” into this sentence. Line 130 is unnecessary

Response:

We appreciate the reviewer’s suggestions to improve readability and style. We have adopted the suggested changes.

Reviewer’s comment:

Line 156 RGGB should be RGB? If not can you define this acronym. Also its something I am unfamiliar with going from RGB to greyscale so it would be useful to have a reference here.

Response:

We are sorry that our use of the RGGB acronym was misleading and that it was confused with the popular RGB acronym. We recognized that the use of the term “channels” also contributed to the misunderstanding. The acronym RGGB refers to the structure of one of the most common Bayer masks which are a set of four colour filters (one red, two green, and one blue) placed in front of the camera’s pixels in order to generate colour images. To prevent confusion, we have decided to remove the use of the phrase “colour channels” and clarify that the raw intensity values of the four pixels covered by the mask were averaged to generate an image with half the pixels height and width of the original image.

Reviewer’s comment:

Line 171 your distortion correction coupled with your FDK recon I am struggling with a bit. With your geometric distortion correction, I assume you are only correcting the issue arising from the camera tilt? If so please say. I assume this since you continue to use FDK for recon which is for circular cone beam. I just want to confirm that your geometrical distortion is not correcting the image effectively to a parallel beam in which case FDK is not the correct algorithm, its just simple FBP. But I think that would introduce more issues. Let me know what is going on here.

Response:

The geometric correction implemented using the square grid corrected only for the geometric distortion caused by the phosphor screen tilt, and for any lens-caused distortion. These corrected projection images were not corrected to a parallel beam. 

Reviewer’s comment:

Line 175 the source alignment in this manner – do you have a reference for this method?

Line 177 “proper alignment of this ring was confirmed”. How? You’re right, its critical so how did you ensure that the ring was perfectly perpendicular to the direction of the source?

Response:

We thank the reviewer for identifying this lack of detail. We have included a suitable reference for the method and specified that the ring alignment was confirmed visually.

Reviewer’s comment:

Line 188 your FDK reconstruction, for a 3104 width detector, you would need 4876 projections according to Nyquist. Its hence a surprise that you would use only 360 projections. How did this decision come about? Can you discuss? You say you then applied x3 binning (calculated going from 39 to 118 micron) because of the spatial resolution… in this case you effectively have a 1034 width detector which requires 1625 projections. It would be an explanation for your poor signal to noise.

I have a feeling this would have been done because of your limit on dose which is fine. Working backwards from Nyquist if you binned by 12 times instead you’ll get really nice reconstructions, albeit at low quality (472 microns). To resolve, I think it would be good to justify your # projections here, then in the discussion talk about additional shielding so you can increase the effective dose and therefore more projections.

Response:

We thank the reviewer for bringing our attention with regards to the number of projections required for sampling our objects of interest following the Nyquist criterion. The decision of having a smaller number of projections than the required was related to scan time, not dose. We believe that since our reconstructions do not show strong sampling-related artifacts and our noise levels are equivalent to other commercially available scanners (running under similar conditions) we can justify this compromise and use it as a strategy to reduce scan time. This strategy has been suggested in one of the papers that the reviewer recommended.

Zwanenburg, E. A., M. A. Williams, and J. M. Warnett. "Review of high-speed imaging with lab-based x-ray computed tomography." Measurement Science and Technology 33, no. 1 (2021): 012003.

We have included text that discusses these considerations in the discussion section of the manuscript. Finally, as discussed in section 4.2 of the previously mentioned publication, it is commonplace to deviate from the strict Nyquist criterion as it is the most impactful way of reducing scan time. 

“The Nyquist sample rule is not met by most operators, indicating that the recommended number of projections can be decreased while not impacting image quality for measurement and observation.”

Figure 13 in the Zwanenburg paper shows that in at least 35 scientific publications reviewed in the study the number of projections used were representative of the parameters used in our study.

Reviewer’s comment:

Throughout this of the different phantoms, I am particularly nervous where you are using dimensional measurements to quantify how good your system is – there is no evidence of a traceable measurement to confirm this against. Have you measured any of them on CMM or SEM or anything? How do you know the volume of porosity in the gyroid to be exactly that (can do an Archimedes test)? The resolution phantom is really nice in concept, but 3D printing structures this small has some issues. Particularly a number of 100um voids that close together. You need to do something to measure these properly and state how this happened.

In the worse case you need to be upfront that none of these have been traceably measured and then redefine their purpose and the subsequent result.

- The porosity phantom. Given a design intent it has been 3D printed. Under this system operation we were able to identify pores as small as X, with an estimated porosity of Y

- The Ti6Al4V I really want a hi-res optical image of the surface with measurements as a minimum. Then you can say at the surface you know these measurements to be true. Then when things are not visible lower down it could be down to the system, or it could be a manufacturing error – you cannot confirm. Further where the line pairs do run further down you must state that you cannot guarantee their size below the surface. Another alternative which might be a nice idea is to scan it with a real X-ray CT system to serve as a gold standard. This would be much higher resolution and provide a sanity check. I am UK based so cannot suggest Canadian locations, but a number of US locations I know about are University Texas has an NSI system with Jessie Maisano where I am sure she would be happy to hear from you, or maybe University of Florida Ed Stanley/Stuart Stock.

Response:

We thank the reviewer for making us realize that the descriptions of our phantoms lacked important details. We agree that providing objective and quantifiable characteristics of these objects is of main importance.

For the case of the CT quality assurance phantom, we were relying on the descriptions from a previous publication about the design of this device (Du et al.). Nevertheless, the reviewer is correct to point out that the information we provided for the phantom included nominal measurements that were lacking information about the precision of the fabrication of the phantom. To solve this doubt, we have modified the nominal values for the inter-bead distances by the average distance between beads measured using an Olympus STM6 measuring microscope with resolution of 0.1 µm. We believe this to be an independent validation of the distances between beads.

With regards to the gyroid-based porous cylinder we apologize for the potential lack of clarity in our writing. But, in the end of section E of the Methods of the paper we describe that the CT measured porosity fraction of this object was compared to the gravimetrically-determined porosity fraction of the 3D printed sample compared to the mass of a solid cylinder printed simultaneously. We also believe that this is an independent validation of the porosity fraction of the object as described in the following publication:

Hong, Greg, Junmin Liu, Santiago F. Cobos, Tina Khazaee, Maria Drangova, and David W. Holdsworth. "Effective magnetic susceptibility of 3D‐printed porous metal scaffolds." Magnetic Resonance in Medicine (2022).

We thank the reviewer for the suggestion to characterize the as-built defects within the titanium phantom. We have taken the opportunity to have the phantom scanned with a commercial scanner (Zeiss Xradia 410 Versa) at much higher energy (150 kVp) and higher resolution (12.6µm). This scan data confirmed that the as-built dimensions differed from the nominal requested design dimensions of the defects (typically built 170µm larger than requested). It also confirmed that the defects were fabricated consistently throughout the phantom. This information has been added to the methods (page, para), results sections (page, para), and discussion.

Reviewer’s comment:

I think section A content is in the wrong place. I realise its an output of the idea which is why you’ve put it where you have, but I feel it should be back in the method with the system description because those choices were clearly already made.

The system design is part of the method, and the results are justifying that this works as you determined.

If not, your method for the system design needs to become less descript in terms of naming components and instead discuss characteristics of the specific components you want. Then Section A in results can cover your choices and why. I think this is messier, so advise you go with my earlier suggestion.

Line 303 – this is the only bit of part A which talks about a result. Given you are moving the rest to the method, I would suggest moving this paragraph into the discussion.

Response:

Although originally, we thought that this section was worth including in the results, we agree that moving these details to the methods section does not affect the description of the project. We have implemented both suggestions.

Reviewer’s comment:

Table 2 – right justify numbers. Also in the caption you should refer to the date these prices were obtained. This will change with time!

Response:

We appreciate the reviewer’s suggestions to improve readability and style. We have adopted the suggested changes.

Reviewer’s comment:

Section C b. needs some work and is confusing. I think what you are trying to say is that after your geometric distortion corrections, the reconstructed volume distances between beads were evaluated against the known values of the beads.

- How were the known values, known? CMM measured? To what accuracy? I doubt they were exactly 35mm, 24.75mm and 49.5mm…

- You jump between measurements of number of voxels and mm. that is confusing. Without going back and doing the maths myself I don’t know that 209 voxels is equal to a particular length we were after!

- It would be useful to have a graph of error vs test length, plotting each distance between the spheres. It should have error bars that represent the 95% confidence interval

Can you please give this short section a rework.

Response:

We thank the reviewer for noticing the possible lack of relevant detail in this section of the manuscript. We have decided to follow the reviewer’s advice and change the reported measurements in mm instead of voxels. And we have included a graph in Figure 11. Showing the comparison between nominal distances and the distances calculated from the CT reconstruction. The graph includes the mean and 95% confidence interval for all the possible distances between all beads.

Reviewer’s comment:

See my comments above about know the precise porosity. The results of the resolution phantom comes across as “look at the figure” and you ran out of steam when writing. I think there is more detail you can put here about the success and not, and how this translates to the limitations of the system. This could be helped by referring to specific sub figures (e.g. 16f) to motivate the discussion, and also demonstrate that the specific figures are actually useful.

Response:

We appreciate the suggestion to elaborate further in the description of the results of the resolution phantom. We have included details with regards to the comparison between the nominal and CT-measured sizes of the defects, as well as commentary with regards to the limitations of the low-cost scanner and critical size of defects.

Reviewer’s comment:

Line 471 This sentence is way too general and actually over reaching. The spatial resolution is one of the key benchmarks of micro-CT systems! So what benchmarks is it the same as?

Response:

We are sorry that the reviewers considered this section over reaching. It was never our intention to claim unjustified characteristics of our low-cost scanner. We have modified the sentence to mention the precise characteristics we were referring to.

Reviewer’s comment:

In fact this entire paragraph is fraught with incorrect info. CMOS in commercial cameras DO have defects, the same as CT detectors. The have “bad pixel” corrections and “noisy pixel” corrections but these are buried in the camera software so you wouldn’t even know they are there. I think I would focus on the benchmarks you broadly say this system exceeds. Quantify numerically. Its probably sensible to quantify the decrease in ability against the cost i.e. this system costs 50x but the spatial resolution is about 10x worse than the highest possible resolution on typical 225kv micro CT scanners, which is 5 um.

Response:

Again, we are sorry to learn our writing was misleading. It was never our intention to claim that our system is better than any other system with regards to fixed-pattern noise and ring artifacts. The intent was to mention the fact that our system was less susceptible to ring artifacts when compared to our pre-clinical scanners, which are equipped with x-ray detectors that use CCD fibre-optic tapered technology. Furthermore, our objective was to mention that due to the specific requirements of the digital photography industry, it is less likely that a high-end digital camera will be sold with critical defects at the detector level. We have modified the text to better communicate these ideas by specifying the detector technology that is more susceptible to ring artifacts, and by specifying that we are only talking about critical defects which are unlikely to be found in commercially available cameras. It is important to clarify that these previously mentioned pre-clinical scanners also implement bad pixel and noisy pixel corrections, but these corrections aren’t enough to prevent ring artifacts in some scenarios. Many of these systems will then include additional ring removal strategies that often compromise image quality. Therefore we do believe that it is an advantage for our proposed detector technology to have less susceptibility to fixed-pattern noise.

Reviewer’s comment:

Line 459. You should easily be able to write a script to better calibrate this. You could do so in matlab and use the Hough transform. Happy to accept the manual approach here but I think you can have a bit more detail on how this can be done.

Response:

We thank the reviewer for their comment with regards to automation of the geometric correction process. We believe that it was out of the scope of this project to design these additional software utilities. We are aware of many different approaches to streamline this specific correction and therefore we believe it is not necessary to specify one specific approach over the other. Furthermore, most of the utilities used for the image reconstruction were developed in C++ for a Linux computer and we do not believe implementations in other platforms such as MatLab are worth exploring.

Reviewer’s comment:

Line 495 When you talk about taking it to site I think it might be good to mention the size case this could theoretically be packed away in and the weight. It will give the idea to the reader of the degree of portability – can I carry in a backpack, or is this a heavier wheel case? Also what power source does it need?

Response:

We appreciate the reviewer’s suggestion and have added information about the size and weight of the hardware in the methods section.

Reviewer’s comment:

Line 499 “as previously mentioned” – remove. I think this first sentence is duplicating what has just been said. Please review how this reads with the previous paragraph talking about archeological sites.

Response:

We appreciate the reviewer’s suggestions to improve readability and style. We have adopted the suggested changes.

---

## [Decision Letter · Decision Letter 1]

23 Sep 2022

Cost-effective micro-CT system for non-destructive testing of titanium 3D printed medical components

PONE-D-21-33035R1

Dear Dr. Cobos,

We’re pleased to inform you that your manuscript has been judged scientifically suitable for publication and will be formally accepted for publication once it meets all outstanding technical requirements.

Kind regards,

Alessandra Giuliani

Academic Editor

PLOS ONE

Additional Editor Comments (optional):

Reviewers' comments:

Reviewer's Responses to Questions

**Comments to the Author**

1. If the authors have adequately addressed your comments raised in a previous round of review and you feel that this manuscript is now acceptable for publication, you may indicate that here to bypass the “Comments to the Author” section, enter your conflict of interest statement in the “Confidential to Editor” section, and submit your "Accept" recommendation.

Reviewer #1: All comments have been addressed

2. Is the manuscript technically sound, and do the data support the conclusions?

Reviewer #1: (No Response)

3. Has the statistical analysis been performed appropriately and rigorously? 

Reviewer #1: (No Response)

4. Have the authors made all data underlying the findings in their manuscript fully available?

Reviewer #1: (No Response)

5. Is the manuscript presented in an intelligible fashion and written in standard English?

Reviewer #1: (No Response)

6. Review Comments to the Author

Reviewer #1: (No Response)

7. PLOS authors have the option to publish the peer review history of their article (what does this mean?). If published, this will include your full peer review and any attached files.

Reviewer #1: No

---

## [Editor Report · Acceptance letter]

29 Sep 2022

PONE-D-21-33035R1 

Cost-effective micro-CT system for non-destructive testing of titanium 3D printed medical components 

Dear Dr. Cobos:

I'm pleased to inform you that your manuscript has been deemed suitable for publication in PLOS ONE. Congratulations! Your manuscript is now with our production department. 

Kind regards, 

on behalf of

Dr. Alessandra Giuliani 

Academic Editor

PLOS ONE